# Atmospheric emissions of hexachlorobutadiene in fine particulate matter from industrial sources

Chenyan Zhao[1,2,3], Lili Yang [1,3] ✉, Yuxiang Sun[1,2,3], Changzhi Chen[1,2,3], Zichun Huang[1,2,3], Qiuting Yang[1,3], Jianghui Yun[1,2,3], Ahsan Habib[4], Guorui Liu [1,2,3] ✉, Minghui Zheng [1,2,3] & Guibin Jiang[1,2,3]

Hexachlorobutadiene (HCBD) is a concerning chemical that is included in the United States Toxic Substances Control Act, and the Stockholm Convention. Knowledge of the sources of HCBD is insufficient and is pivotal for accurate inventory and implementing global action. In this study, unintentional HCBD release and source emission factors of 121 full-scale industrial plants from 12 industries are investigated. Secondary copper smelting, electric arc furnace steelmaking, and hazardous waste incineration show potential for large emission reductions, which are found of high HCBD emission concentrations of > 20 ng/g in fine particulate matter in this study. The highest HCBD emission concentration is observed for the secondary copper smelting industry (average: 1380 ng/g). Source emission factors of HCBD for the 12 industries range from 0.008 kg/t for coal fire power plants to 0.680 kg/t for secondary lead smelting, from which an estimation of approximately 8452.8 g HCBD emissions annually worldwide achieved. The carcinogenic risks caused by HCBD emissions from countries and regions with intensive 12 industrial sources are 1.0-80 times higher than that without these industries. These results will be useful for formulating effective strategies of HCBD control.

Global commercial production of hexachlorobutadiene (HCBD) was prohibited by the Stockholm Convention in 2015[1] because of HCBD's high toxicity, persistence, and potential for bioaccumulation and long-range atmospheric transport. Further controls on unintentional emissions were implemented in Annex C of the Stockholm Convention updated in 2017[2]. Despite worldwide efforts to address global HCBD accumulation, it is ubiquitous in the environment worldwide, including in the atmosphere, water, and soil[3–10].

HCBD has been recognized as the most nephrotoxic aliphatic chlorinated hydrocarbon in 1978, capable of inducing epithelial necrotizing nephritis[11]. The occurrences of HCBD in the atmosphere may potentially lead to respiratory damage and carcinogenic effects.

Because of its relatively high volatility and long half-life, HCBD easily stays in the atmosphere[12]. During a test conducted by the United Nations Environment Programme (UNEP), atmospheric measurements above Mongolia revealed an exceptionally high concentration of 344 ng/PUF (polyurethane foam)[7]. This concentration far surpassed those of hexachlorobenzene, polychlorinated biphenyls, and other chlorinated organic pollutants in the same samples, and exceeded dichlorodiphenyltrichloroethane concentrations in the same area by two orders of magnitude. This persistence underscores the need for ongoing monitoring and mitigation strategies to curb the impact of HCBD on ecosystems and human health. The lack of knowledge of HCBD sources makes source apportionment of HCBD in the

[1]State Key Laboratory of Environmental Chemistry and Ecotoxicology, Research Center for Eco-Environmental Sciences, Chinese Academy of Sciences, Beijing, China. [2]School of Environment, Hangzhou Institute for Advanced Study, UCAS, Hangzhou, China. [3]College of Resources and Environment, University of Chinese Academy of Sciences, Beijing, China. [4]Department of Chemistry, Dhaka University, Dhaka, Bangladesh. ✉e-mail: llyang@rcees.ac.cn; grliu@rcees.ac.cn

environment challenging. To date, the sources leading to the high concentrations of HCBD in Mongolia are unknown. Moreover, because it undergoes extensive long-range atmospheric transport, HCBD has been found in areas remote from human industrial activities, including Arctic soil[13] and Antarctic ice cores[14].

Manufacture of HCBD as a commercial chemical and unintentional release from industrial activities are two sources of atmospheric HCBD pollution. Intentional production of HCBD as a commercial chemical will gradually stop under the Stockholm Convention, and unintentional release of HCBD from industrial activities will become a more important source. Research suggests that as industrial output increases, HCBD emissions are expected to rise in the future[15]. Currently, a few studies have reported the release of HCBD from chemical production processes that use chlorine and waste disposal[16,17]. Our previous study has also investigated the occurrence of HCBD in products and bottom liquid of chlorobenzene, trichloroethylene, and tetrachloroethylene chemical manufacturing plants (Supplementary Table 1)[16]. HCBD concentrations in the bottom liquid samples contributed 24%-99% of the total HCBD formed in the chemical production plants. The bottom liquid was disposed of as hazardous waste by incineration[18]. Therefore, a proportion of HCBD from commercial chemical manufacturing processes would finally enter into environment by the unintentional releases from incinerations of bottom liquid. However, current HCBD emission data from industrial sources were very limited. It has also been found that HCBD occurred in industrial fine particulate matter (i-PM) from waste incineration facilities in East China[19]. Most i-PM were intercepted by air pollution control devices, a small fraction is released directly into the atmosphere. Those i-PM size were characterized less than 2.5 μm by scanning electron microscopy in our previous study[20]. The HCBD adsorbed on $PM_{2.5}$ (PM with a diameter of <2.5 μm)[21] can readily be inhaled and pose health risks to humans[22].

Although industrial activity is prolific, there has been little research on HCBD sources and the available data are outdated. Therefore, it is challenging to accurately quantify the emission of HCBD from industrial activity. Identifying potential emission sources and accurately quantifying HCBD released from various industrial sources is key for effective regulation of unintentional emissions of HCBD. The emission factor (EF) methodology recommended by the European Monitoring and Assessment Programme of the European Economic Area[23] and the UNEP[24] is usually used to compile emission inventories. However, inventory compilation has not been conducted for HCBD because of the lack of EFs.

In this study, 121 full-scale industrial plants from 12 industries are investigated for HCBD emissions. The selected 12 industries, which are all important sources of unintentional persistent organic pollutants (POPs), are secondary zinc smelting (SZn), secondary lead smelting (SPb), secondary copper smelting (SCu), secondary aluminum smelting (SAl), primary copper smelting (PCu), hazardous waste incineration (HWI), municipal solid waste incineration (MWI), iron ore sintering (IOS), electric arc furnace (EAF) steelmaking, coke production (COP), cement kiln (CK) co-processing, and coal-fired power plants (CFPP). EFs for these 12 industries are derived according to the UNEP methodology, which will be essential data for compiling a global emission inventory[24]. Here, emission levels and EFs of HCBD from different industries are compared, offering a comprehensive benchmark for recognizing the priority-controlled sources of unintentional HCBD. To understand the potential impact of industrial emissions on human burdens and health risks, we map the spatial distribution of HCBD emissions on a national scale and estimate the potential carcinogenic risk of HCBD, using the human health risk assessment manual[25]. This study clarifies the HCBD unintentional emission inventory globally from industrial sources and will be helpful for formulating effective international strategies of HCBD control.

## Results and discussion
### Comparison of HCBD concentrations in PM from multiple industrial thermal sources

There is a lack of monitoring data for HCBD from industrial emissions. Some previous studies have reported the concentrations of HCBD in water bodies[3,5,26,27] and polluted land[3,9] near sewage treatment plants. However, there is currently no direct evidence to support the direct emission of HCBD from industrial sources into the environment. Monitoring and comparison of HCBD concentrations in particulate matter from various industrial processes can aid in identifying industries that show potential for emission reductions. Unintentional HCBD release and source emission factors for 12 industries are derived in this study.

HCBD was detected in 86% of the 121 samples from the 12 industries (Supplementary Table 2). The highest HCBD concentration was found in the i-PM samples from SCu (1379.6 ng/g). The HCBD concentrations in samples from the other 12 industries ranged from 2.0 to 37.2 ng/g. Emissions from EAF steelmaking and HWI had relatively high HCBD concentrations at 37.2 and 23.3 ng/g, respectively. The concentrations in emissions from COP, CFPP, and SAl were comparable at 5.6, 6.7, and 5.1 ng/g, respectively. The concentrations in emissions from PCu, SPb, CK co-processing, MWI and SZn were <5 ng/g, much lower than that from the other industries.

Our results show that, in addition to large amounts of particulate matter and carbon dioxide[28], CFPP emit high concentrations of HCBD in the particulate matter. Variations in emissions were apparent among different CFPP, and the HCBD concentration ranged from 0.1 to 46.3 ng/g. High concentrations of HCBD were also observed for samples from EAF steelmaking, with an average concentration of 37.2 ng/g. Among the EAFs, EAF-2 had a HCBD concentration of 441.6 ng/g, which was the next highest concentration after SCu. This high concentration of HCBD in EAF steelmaking could be caused by the presence of organic residues in scrap raw materials, such as rubber, cable wrapping, plastics, and polyvinyl chloride, which provide additional sources of chlorine for the unintentional formation and emission of HCBD[29]. Yang et al. speculated that differences in the chlorine contents of raw materials and metal catalysts are important contributors to differences in the emissions of chlorinated organic pollutants[30,31]. In conjunction with the results from previous research, the large variation in HCBD concentrations among the samples could be attributed to differences in the steelmaking process. An earlier study suggested that EAF steelmaking that incorporates preheating raw material stages may produce higher concentrations of chlorinated pollutants[30].

COP has long been recognized as an important source of unintentionally produced POPs. Dioxins, polychlorinated biphenyls, and polychlorinated naphthalenes have been detected in i-PM from COP in previous studies[32–35]. In the present study, HCBD was detected in 11 out of the 13 samples obtained from COP at concentrations ranging from 0.6 to 19.0 ng/g (average: 5.6 ng/g). MWI is considered an important source of HCBD[17,19,36]. The higher concentrations of HCBD from COP than MWI suggest that COP should be considered as an important source.

For CK co-processing, HCBD was detected at 14.9 ng/g. Co-processing in a CK is a common method for solid waste treatment. The highest temperature in this process can exceed 1300 °C, which effectively destroys waste residues containing HCBD and other chemicals[37,38]. The HCBD concentrations in the i-PM in the present study were much higher than those detected in previous co-incineration experiments (0.68 and 0.25 ng/g)[18]. This indicates that HCBD is generated during CK co-processing, and variation in HCBD concentrations in i-PM samples could be caused by differences in raw materials and temperature[39].

The HCBD concentration in i-PM from the HWI was higher than that in i-PM from MWI. This could be attributed to the fact that, HCBD-containing waste has been classified and managed as hazardous waste

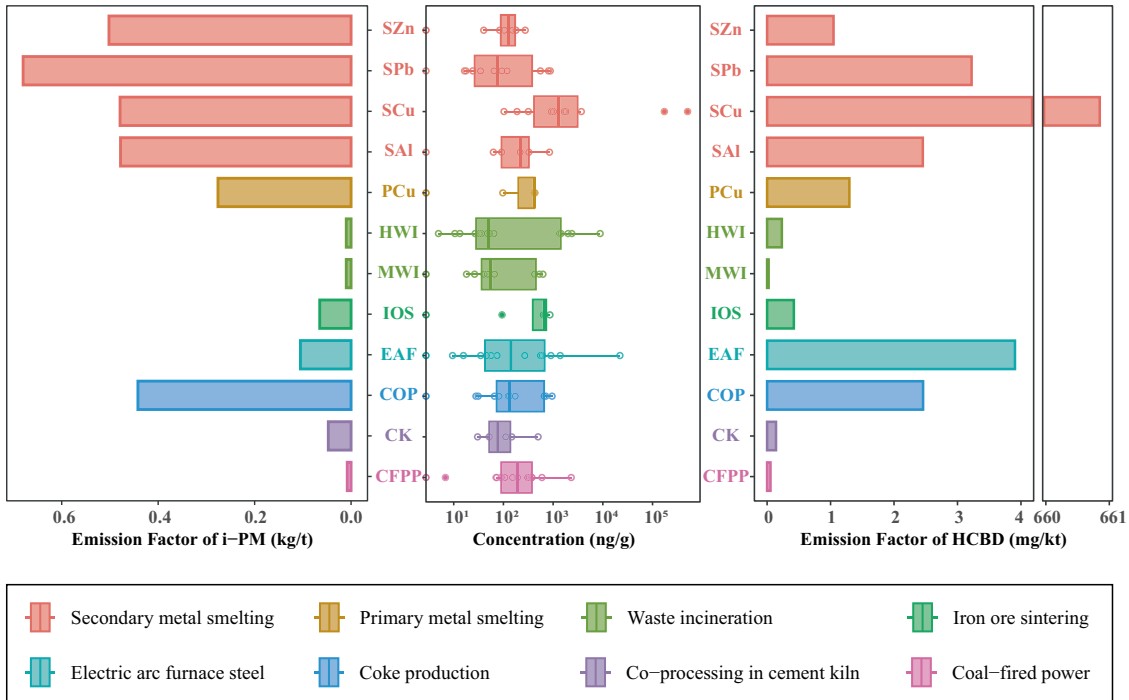

**Fig. 1 | Concentrations of hexachlorobutadiene (HCBD) in industrial fine particulate matter (i‑PM) from 12 industries; Emission factors of i‑PM and emission factors of HCBD in i‑PM from 12 industries.** SZn, secondary zinc smelting; SPb, secondary lead smelting; SCu, secondary copper smelting; SAl, secondary aluminum smelting; PCu, primary copper smelting; HWI, hazardous waste incineration; MWI, municipal solid waste incineration; IOS, iron ore sintering; EAF, electric arc furnace steelmaking; COP, coke production; CK, cement kiln co-processing; CFPP, coal-fired power plants.

since it was added to the Stockholm Convention. Fly ash from waste incineration facilities can cause environmental pollution through leachate and surface soil contamination[36]. This underscores the importance of the hazardous waste treatment industry for controlling HCBD[40]. In previous study, the direct quantification of HCBD from industrial emission sources is only available for waste incineration sources[41]. HCBD was detected in fly ash and flue gas samples obtained from industrial waste incineration, exhibiting concentrations of 0.25 ng/g dw and 8.2 ng/Nm³, respectively[18]. The study conducted by Zhang et al. have similar findings, with the concentration of HCBD in municipal solid waste incineration ranging from 1.39 to 7.81 ng/g, while hazardous waste incineration exhibited a higher concentration of 43.8 ng/g[19]. Those results are in the same order of magnitude with data for waste incineration sources.

## Atmospheric EFs and inventory of HCBD from industrial activities

Unlike HCBD emissions from the chemical industry, such as its release from products during use[42], emissions of HCBD from industrial thermal sources are frequently directly released into the atmosphere and lead to environmental pollution[15]. Currently, there is very limited research on the atmospheric content of HCBD, with only a few studies primarily focusing on HCBD concentrations in industrialized regions. The urban atmosphere, characterized by a high concentration of petroleum enterprises[43], has been found to contain HCBD at levels ranging from 4.98 to 20.20 µg/m³. In two studies conducted in 2017 and 2018 near organochlorine pesticide factories, elevated concentrations of HCBD were detected, with levels ranging from 0.03 to 0.33 ng/m³ and from 0.01 to 2.23 ng/m³, respectively. In 2018, HCBD concentrations of 0.21 µg/m³ was detected near a chlor-alkali plant in Catalonia, Spain[44]. EFs are important for estimating pollutant discharge[45]. To date, the EFs of HCBD from various industries have not been comprehensively studied. EFs can be combined with historical usage data

to identify high emission levels and contributing industries. The geometric mean EFs for the 12 industries in the present study are shown in Fig. 1.

Among the investigated industries, SCu had the highest EF for HCBD. This was attributed to the very high concentrations of HCBD in i‑PM from this industry and the high EF of i‑PM (0.48 kg/t). The EF of HCBD from SCu was one order of magnitude higher than that of polychlorinated naphthalenes and other POPs released during SCu[46]. Therefore, the atmospheric emission of HCBD from SCu should receive attention. COP also had a high HCBD EF. The EF of HCBD from COP was one order of magnitude higher than the EFs for pentachlorobenzene and hexachlorobenzene in a previous study[47]. The EFs for SPb, SAl, and EAF steelmaking were comparable to those for COP. These results suggest that the above industries are important HCBD sources, and that exposure of workers to HCBD in these industries might be high.

Because HCBD is persistent, HCBD adsorbed on i‑PM will stay in the atmosphere for a long period after release. Most of the HCBD emitted undergoes long-range atmospheric transport, resulting in atmosphere pollution on a regional scale[48]. Therefore, it is important to calculate the atmospheric release of HCBD from industrial sources. The EFs of HCBD from different sources were derived from the monitoring data for industrial plants according to methodology recommended by the European Monitoring and Evaluation Programme[23] of the European Economic Area and the UNEP[24]. The EFs and activity levels of the investigated industries were used to calculate that China unintentionally emitted approximately 3636.0 g of HCBD into the atmosphere. Among the industries, SCu accounted for the largest proportion of HCBD emissions (41.8% of the total). This was followed by COP with a contribution of 29.6% to the total emissions of HCBD. EAF steelmaking, CFPP and the CK co-processing also had large contributions to the total HCBD emissions at 10.0%, 9.2%, and 8.5%, respectively. Together, the above five industries contributed to 99.1%

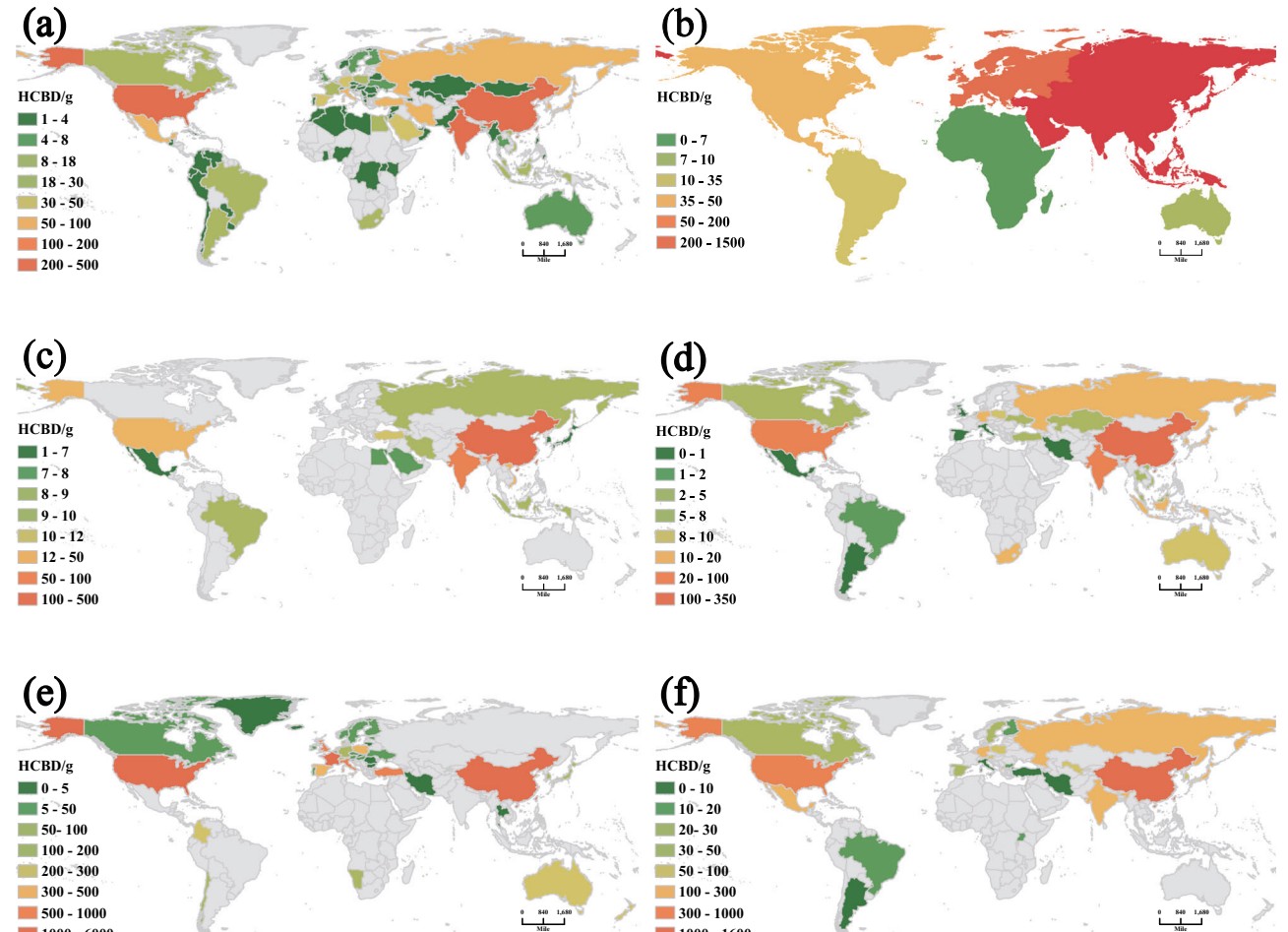

**Fig. 2 | Estimates of hexachlorobutadiene (HCBD) emissions from various industries in different countries and regions in 2018. a** Electric arc furnace steelmaking, **b** coke production, c cement kiln co-processing, **d** coal-fired power plants, **e** municipal solid waste incineration, and **f** secondary copper smelting. We draw the map according to vector data sourced from "Database of Global Administrative Areas free vector data" (https://gadm.org/) by Arcgis Pro software.

of the total emissions from the 12 investigated industries. The spatial distributions of HCBD in i-PM from industrial activity in different provinces of China are shown in the Supplementary Fig. 1.

Next, we estimated the global HCBD emissions in i-PM from the identified six priority industries (SCu, EAF steelmaking, COP, CFPP, CK co-processing, and MWI) in world because these six industries together contributed > 98% of the total HCBD emissions from investigated 12 industries.

Figure 2 shows HCBD emissions from different countries and regions across 6 controlled industries. Among the emission sources, SCu was the largest with emission of approximately 2643.3 g of HCBD. This was attributed to the extremely high concentration of HCBD in i-PM from this industry. China, the US, and Japan are the main contributors to HCBD atmospheric emissions from SCu. The countries with high HCBD atmospheric emissions are mainly high-income countries, which could be attributed to the high proportion of recycled copper used in production of refined copper in these countries. For instance, recycled copper is used fo40% of SCu in France, and >80% of SCu in Germany and Italy. EAF steelmaking emitted approximately 2043.2 g of HCBD in 2018. As a traditional steelmaking technology, the global annual output of steel from EAF was 523,142 kt in 2018. China, India, and the US were the main contributors to the HCBD atmospheric emissions for this industry. South Korea, Italy, and Turkey had the highest per capita emissions. South America had the lowest HCBD atmospheric emissions for EAF steelmaking. In 2018, COP

released 1667.5 g of HCBD into the atmosphere. Although the HCBD concentration in the i-PM from COP was not high, the emission factor of the i-PM was high. COP is an important metallurgical industry and the world output was 683,000 t in 2018. Coke is the raw material for iron and steelmaking, and can be present in the products of these industries and transferred to other industrial processes. Therefore, COP is an important source of HCBD atmospheric emissions. COP has multiple processes that emit i-PM[20], including coal loading, coke discharge, and coke oven heating, and its i-PM EF is large. Asia has high levels of COP, and China, as a major producer and exporter of coke, accounts for more than 50% of the world's annual COP. The COP industry in Europe released 192 g of HCBD in 2018, which was higher than the total annual release for SZn, SPb, SAl, and PCu. As important waste disposal methods, MWI and CK co-processing released 742.9 and 569.2 g of HCBD in 2018, respectively. MWI and CK co-processing have a high correlation with population density because they are closely related to the production of domestic waste and hazardous waste. This is mainly reflected in the fact that the per capita HCBD emissions of each country are relatively similar. The HCBD emissions from CFPP, SAl, SZn, and SPb were all below 100 g/year.

Figure 3 shows the global atmospheric emissions of HCBD in 2018. A total of 101 countries and regions emitted 8452.8 g of HCBD in industrial i-PM. With the continuous growth of the global population, urbanization is accelerating, leading to a gradual increase in population density. Simultaneously, as population density rises, so does

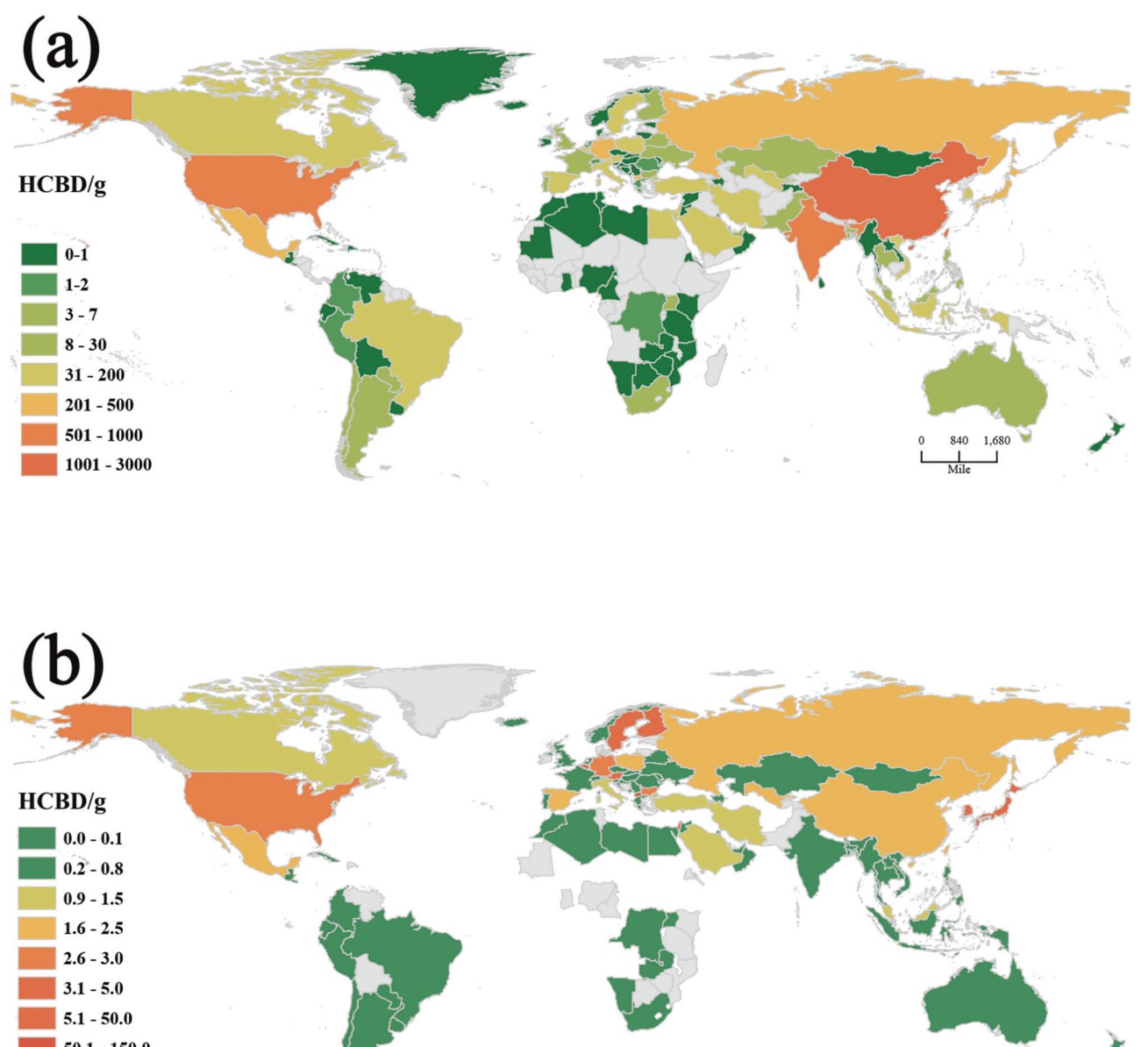

**Fig. 3 | Global hexachlorobutadiene (HCBD) atmospheric emissions in 2018. a** total release, **b** per capita emissions. We draw the map according to vector data sourced from "Database of Global Administrative Areas free vector data" (https://gadm.org/) by Arcgis Pro software.

industrial demand. We link HCBD emissions with population data and calculate per capita emissions of HCBD in different countries. Higher per capita emissions are primarily observed in countries with high income, which were listed by the World Bank (2018), such as Belgium (9.8 µg/per), Australia (8.8 µg/per), Sweden (4.2 µg/per), and South Korea (3.3 µg/per). The emissions of POPs are significantly influenced by the industrial capacity of different countries[49,50]. The level of industrial production, particularly in countries with high income, is closely associated with economic development. Therefore, economic indicators, such as gross domestic product (GDP), display substantial correlations with pollutant emissions. On a global scale, we observed a significant positive correlation between HCBD emissions (in grams) and the national normalized GDP (in billions of dollars) in both the Global South and Global North (Fig. 4). This suggests that human socioeconomic activities have marked contributions to HCBD emissions. The linear regression plot of the relationship between HCBD

emissions (in grams) and GDP (in billions of dollars) shows data for 98 of countries that had HCBD emissions in 2018; four countries were excluded because of a lack of GDP data.

Global South ($r = 0.98$) exhibited a higher correlation with HCBD than Global North ($r = 0.88$). The higher correlation between HCBD emissions and GDP in the Global South may be explained by the following: (1) secondary industries (including manufacturing, mining, and construction) have larger contributions to GDP in the Global South than in the Global North[51]; and (2) many countries in the Global North have completed more comprehensive infrastructure construction in the past, while countries in the Global South are in a stage of rapid development accompanied by rapid GDP growth[52,53]. Higher emissions of HCBD in the Global South than in the Global North indicate that individuals in the Global South may have higher exposure to HCBD than in the Global North. In addition, considering the completely different climate conditions in the Global South and Global North, the

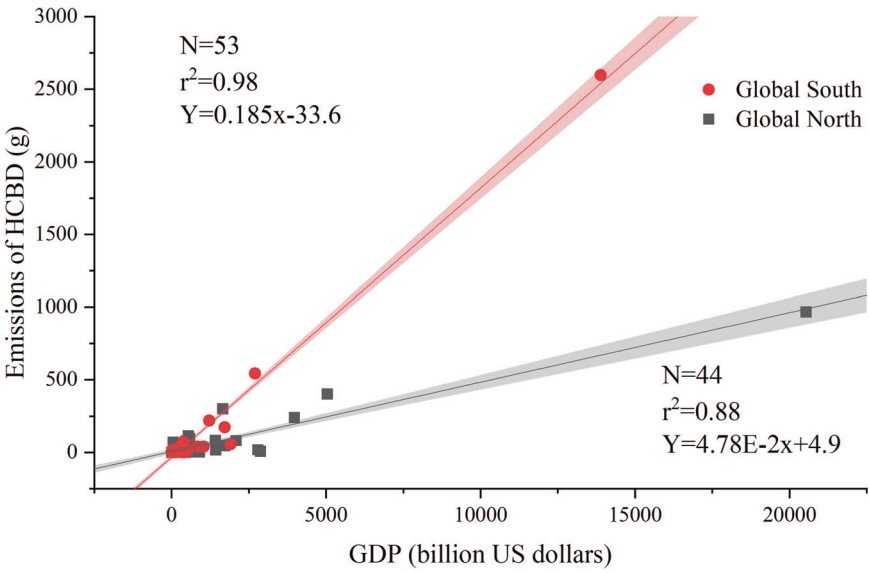

**Fig. 4 | Linear regression plot of the relationship between hexachlorobutadiene (HCBD) emissions and national normalized gross domestic product (GDP) in 98 countries.**

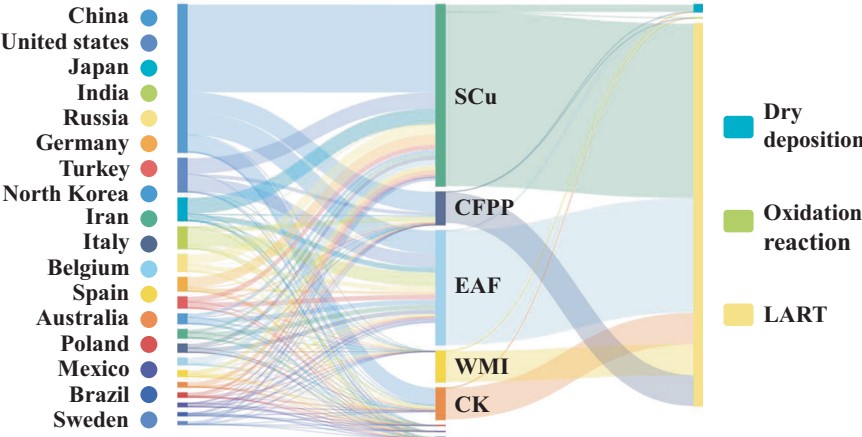

**Fig. 5 | Consumption pathways (dry deposition, oxidation reactions, long-range atmospheric transport [LART]) of hexachlorobutadiene (HCBD) in particulate matter released from different industries.** SCu secondary copper smelting, CFPP coal-fired power plants, EAF electric arc furnace steelmaking, MWI municipal solid waste incineration, CK cement kiln co-processing.

environmental behavior of HCBD, such as dry deposition, oxidation reactions, were different and should be taken into consideration when comprehensively assessing HCBD exposure risks in these regions.

Our results show that these 12 thermal industries are important sources of HCBD, but the existing policies lack the regulation of HCBD in thermal industrial processes. In China, HCBD appears to be primarily limited to industries such as petroleum refining. Therefore, it is imperative to investigate the formation mechanism across diverse industrial processes and strengthen emission limits pertaining to HCBD from various industrial sources. Here, 12 types of industrial sources were analyzed for HCBD release. It provides a data basis for the formulation of HCBD emission inventory and a scientific basis for global control and emission reduction of HCBD.

This study identified the priority control industries SCu, EAF and COP for high HCBD emissions. Optimizing the technical processes of the above industries and focusing on strengthening terminal emission control can effectively reduce the global emissions of HCBD and the potential health risks caused by HCBD.

## Behavior of HCBD in the atmosphere and the associated carcinogenic risk

Atmospheric transport plays an important role in facilitating the movement of substances like HCBD from their sources to remote areas, and contributes to environmental contamination[14,54–56]. Once HCBD enters the atmosphere, it can undergo different processes, such as degradation by reaction with free radicals in the air, and enter soil and water environments through processes like dry deposition and wet deposition[57–59]. It is crucial to calculate the amount of HCBD that will undergo long-range atmospheric transport after emission from the source.

The proportions of HCBD lost to dry deposition or consumed by reactions in the atmosphere near the source and undergoing long-range atmospheric transport away from the source were similar for different industries (Fig. 5). The annual emission rate of HCBD was 8452.8 g/year. Of this, approximately 3% (248 g/year) was lost through dry deposition, and less than 1% was removed by oxidation reactions in the atmosphere.

These values show that only a small fraction (<4%) of HCBD emitted into the atmosphere is lost to dry deposition and oxidation reactions near the emission source. More than 96% of the HCBD released from industrial sources persists and undergoes long-range atmospheric transport. The transported HCBD may accumulate in high-altitude areas with low temperatures and high precipitation, and this could result in an alpine condensation effect and widespread regional and global pollution[54,60–62]. Because of the long-range atmospheric transportation, HCBD is detected in high-altitude areas distant from high-intensity human activities[59,63].

Worldwide, atmospheric $PM_{2.5}$ exposure leads to significant increases in annual premature deaths and carcinogenic risk[64–66]. Previous research has demonstrated that 47% of the current global $PM_{2.5}$-related deaths can be attributed to air pollution from traditional industrial combustion sources, such as fossil fuel combustion[65,67]. However, few studies have investigated the contribution of HCBD in $PM_{2.5}$ to carcinogenic risk. In the present study, we evaluated the potential carcinogenic risk of HCBD in industrial $PM_{2.5}$ using the methods recommended by the US Environmental Protection Agency (EPA) in their manual for human health risk assessment.

Supplementary Tables 3 and 4 lists the carcinogenic risks of HCBD emitted into the atmosphere to the general population worldwide. The carcinogenic risks caused by HCBD emissions from the 12 industrial sources were between 0.005% and 7930%. The carcinogenic risk of HCBD inhalation for children ($8.7 \times 10^{-7}$ to $7.0 \times 10^{-5}$) was higher than that for adults ($5.5 \times 10^{-7}$ to $4.43 \times 10^{-5}$). The results were divided into two categories (may cause a cancer risk or will not cause a cancer risk), and the threshold was the recommended carcinogenic risk value of $10^{-6}$ from the US EPA human health risk assessment manual[68]. Among the countries and regions, 51.49% had carcinogenic risk for air inhalation of HCBD. Ultra-high concentrations of HCBD detected in Mongolia[7] and the Pacific islands[6] by the UNEP also suggest that there are high exposure and health risks in certain regions. In the present study, the HCBD emissions from the 12 industries sled to an increase of $6.81 \times 10^{-11}$ to $1.13 \times 10^{-4}$ in the carcinogenic risk. We also calculated the carcinogenic risk for HCBD inhalation in the general population in different regions. The results showed that increased inhalation of HCBD because of $PM_{2.5}$ emissions from the 12 industries occurred in Europe, Oceania, the Middle East, Central and South America, Asia, and Africa, and was associated with different degrees of potential carcinogenic risk.

Among the investigated 12 industrial sources, SCu, EAF, and HWI, had HCBD concentrations > 20 ng/g in i-PM and are potential priority sources for HCBD control and regulations. The EFs for the 12 industries were derived to provide data for establishing an HCBD emission inventory. The atmospheric HCBD emission concentrations in PM from these industries globally were assessed.

The dry deposition rates and oxidation reaction rates of HCBD released from the 12 industries were 270 and 0.1 g/year, respectively. Over 96% of HCBD is released into the atmosphere along with $PM_{2.5}$ and undergoes long-range transport. HCBD atmospheric emissions may pose risks in certain countries and regions because of intensive industrial activity. Consequently, it is essential to implement effective measures aimed at reducing emissions and human exposure.

## Methods

### Sample information
Methodology flowchart of this study was shown in Supplementary Fig. 2. Currently, the main source of HCBD is unintentional emissions from industrial processes because the production and use of HCBD have been banned under the Stockholm Convention. Unintentionally released HCBD might have similar sources to polychlorinated dibenzo-p-dioxins and dibenzofurans. According to the UNEP toolkit (2013), we focused on 12 industries: SZn, SPb, SCu, SAl, PCu, HWI, MWI, IOS, EAF steelmaking, COP, CK co-processing of waste, and CFPP. We selected 6-15 plants for each industry, which is much larger than the numbers of plants recommended in the UNEP toolkit (2013)[24].

Samples of i-PM were collected at 121 full-scale plants in the 12 different industries (Supplementary Fig. 3). The industrial plants were 13 for COP (COP-1 to COP-13), 15 for CFPP (CFPP-1 to CFPP-15), 14 for EAF steelmaking (EAF-1 to EAF-14), 6 for CK co-processing (CK-1 to CK-6), 11 for MWI (MWI-1 to MWI-11), 14 for HWI (HWI-1 to HWI-14), 6 for PCu (PCu-1 to PCu-6), 10 for SCu (SCu-1 to SCu-10), 6 for SAl (SAl-1 to SAl-6), 8 for SZn (SZn-1 to SZn-8), 11 for SPb (SPb-1 to SPb-11), and 7 for IOS (IOS-1 to IOS-7).

I-PM samples were collected from dust collectors that had been operating for at least 72 h. The dust collectors were bag filters or electrostatic precipitators, which are widely used as air pollution control devices and can filter 99% of the particulate matter in the metal smelting process to reduce the release of HCBD to the atmosphere. The samples were used to represent the average concentration for each process. EAF steelmaking samples were collected at the ash funnel of the bag filter in the preheating stage and smelting stage of scrap steel. The samples were collected with stainless steel spoons, packed in aluminum foil, and sealed in polypropylene bags. They were then immediately transported to the laboratory and stored at a low temperature in the dark.

### Chemicals and reagents
Solvents (acetone, dichloromethane, and n-hexane; pesticide residue analysis grade) were acquired from Avantor Performance Materials (Allentown, PA). Silica gel and anhydrous sodium sulfate were baked at 550 °C for 6 h before use. Florisil (60-100 mesh) was baked at 550 °C for 12 h before use. $^{13}$C-labeled HCBD was provided by Cambridge Isotope Laboratories (Andover, MA).

### Sample pretreatment and instrumental analysis
We summarized the extraction and purification methods for HCBD determination in solid samples in recent years (Supplementary Table 5). In order to ensure that HCBD is extracted completely, each sample was extracted with acetone/n-hexane (1:1, v/v) by Soxhlet extraction for 12 h and then reduced to 1 mL by rotary evaporation. A multilayer column comprising 2 g of silica gel, 4 g of Florisil, and 4 g of silica gel (from bottom to top) was selected as the optimal clean-up method due to its superior purification efficiency. Isotopic dilution gas chromatography-mass spectrometry techniques (GC-MS) are the internationally recognized optimum method for trace analysis of POPs in complex environmental matrices. Many international analytical standards, such as US EPA method TO-13A and ISO 23646, recommended and used isotopic dilution GC-MS techniques. In our study, for more precise quantification, we also adopted isotopic dilution GC-MS techniques for HCBD quantification. Prior to sample pretreatment, $^{13}$C-labeled HCBD used as internal standard was spiked into the samples. $^{13}$C-labeled HCBD standard was utilized for accurate quantitation of HCBD in samples. The HCBD concentration in the sample was determined by GC-MS (7890 A GC/5975 MSD; Agilent Technologies, Santa Clara, CA) using an HP-5MS column (30 m × 0.25 mm i.d., 0.25 µm film thickness; Agilent Technologies). A standard solution (1 mg/mL) was analyzed by GC-MS in full scan mode. The standard spectrum was used to identify appropriate quantitation and confirmation ions (Supplementary Fig. 4).

### Quality control
To ensure the accuracy of the experimental data, we implemented rigorous quality assurance and quality control procedures throughout the sample collection, cleanup, and analytical processes. These procedures included the simultaneous collection, cleanup, and analysis of field blanks. We analyzed the entire sample using a validated highly sensitive and stable experimental method and conducted program blank experiments for each sample batch.

**Table 1 | Dry deposition velocity of acenaphthene**

| Area | Dry deposition velocity (cm/s) | Location | Elevation (m) | Reference |
|---|---|---|---|---|
| Bursa, Turkey | 0.7 | 30°00′N, 40°40′E | 100 | 57 |
| Florida, USA | 0.15 | 27°53′N, 82°32′W | 5 | 78 |
| Beijing, Tianjin, China | 0.006 | | 44 | 79 |
| Brno, Czech Republic | 0.05-0.2 | 49°10′N, 16°34′E | 308 | 80 |
| California, USA | 0.2 | 33°45′N, 118°16′W | 100 | 81 |

**Emission estimation**

To estimate the HCBD emissions, we combined different emission sectors and EFs to establish data sets. We used the following equation to estimate the HCBD emissions:

$$E = \bar{C}_l \times EF_i \times Y_i, \tag{1}$$

where $E$ is the annual emission of HCBD in each industry; $\bar{C}_l$ is the average value of the HCBD concentration for each plant in the same industry; and $EF_i$ and $Y_i$ are the average EF of i-PM (Supplementary Table 6) and annual production yield in the specific industry (Supplementary Table 7), respectively. The $Y_i$ data were collected from relevant international statistics organization. We multiplied the industrial activity levels by the fraction of each emission sector to obtain the emission inventory for each emission sector.

Dry deposition ($D_p$) is one way for HCBD to pollute the area around an industrial plant. Equation 2 was used to estimate the $D_p$ of HCBD around the plants in this study.

$$D_p = C_i \times C_{PM2.5} \times V_d \times S, \tag{2}$$

where $C_i$ is the average concentration (ng g$^{-1}$) of HCBD in i-PM emitted by all plants in different industries; $C_{PM2.5}$ is the concentration of PM$_{2.5}$ in the plant area (µg m$^{-3}$); $V_d$ is the dry deposition velocity (cm·s$^{-1}$) for HCBD; and $S$ represents the area of HCBD affected by dry deposition under natural conditions, which is the geographical range or spatial coverage of HCBD deposition within a specified time. To date, no accurate data have been reported for the HCBD dry deposition velocity. Acenaphthene has a boiling point close to that of HCBD, and its dry deposition data are summarized in Table 1. The dry deposition velocity in urban residential areas is typically between 0.006 to 0.2 cm s$^{-1}$. However, it is important to note that in industrial areas, the dry deposition velocity tends to be higher, as observed in the study by Esen et al. (0.7 cm·s$^{-1}$)[57]. This higher rate in industrial areas can be attributed to many factors, with the heat island effect likely playing an important role.

The dry deposition coefficient ($m$) is the ratio of the dry deposition to the yield of the individual industrial plant. Therefore, the dry deposition rates (D$_p$) of different industries can be calculated using Eqs. 3 and 4:

$$m = \frac{D_p}{Y_0}, \tag{3}$$

$$D_{p.i} = m \times Y_i = \frac{D_p}{Y_0} \times Y_i, \tag{4}$$

where $Y_i$ is the total output of industry $i$, and $Y_0$ is the annual output of the individual plant.

The oxidation rate ($O_x$) of HCBD was calculated using Eq. 5.

$$O_x = \sum K_i \times C_i \times C_{PM2.5} \times C_o \times V_s, \tag{5}$$

where $K_i$ is the rate constant for the reaction between HCBD and oxidant $i$ (cm$^3$ mol$^{-1}$ s$^{-1}$), $C_i$ is the average concentration (ng g$^{-1}$) of HCBD in

**Table 2 | The reaction rate constants of different oxidants with hexachlorobutadiene and their contribution to the total oxidation amount**

| Oxidants | Hydroxyl radical | Chlorine radical | Nitrogen tri-oxide radical | Hydrogen peroxide |
|---|---|---|---|---|
| $K_i$ (cm$^3$ mol$^{-1}$ s$^{-1}$)[59] | $6.33 \times 10^{-16}$ | $4.51 \times 10^{-13}$ | $1.32 \times 10^{-20}$ | $4.33 \times 10^{-29}$ |
| $C_o$ (mol cm$^{-3}$)[59] | $1.4 \times 10^6$ | $9.0 \times 10^3$ | $4.81 \times 10^7$ | $1.2 \times 10^8$ |
| Contribution (%) | 17.92 | 82.08 | $1.07 \times 10^{-3}$ | $6.59 \times 10^{-6}$ |

$K_i$ rate constant for the reaction; $C_o$ concentration of oxidant.

i-PM emissions from all plants across different industries, $C_{PM2.5}$ is the concentration of PM$_{2.5}$ in the plant (µg m$^{-3}$), $C_o$ is the maximum concentration of oxidant $i$ (mol cm$^{-3}$), and $V_s$ is the volume of the plant (km$^3$). The simulated results have shown in Table 2.

Long-range atmospheric transport of HCBD ($T_l$) was calculated as Eq. 6:

$$T_1 = E - D_p - O_x. \tag{6}$$

where $E$ is the emissions amount of HCBD from each industry.

**Cancer risk assessment for air inhalation**

The intake factor and carcinogenic risk for HCBD respiratory exposure in the general population were calculated using Eqs. 7 and 8.

$$IF = \frac{IR \times ET \times Abs \times EF \times ED}{BW \times AT}, \tag{7}$$

$$CR = C_i \times IF \times CSF, \tag{8}$$

where $C_i$ is the concentration of pollutant $i$ (pg·m$^{-3}$), IF is the intake factor for children or adults (m$^3$·kg$^{-1}$d$^{-1}$), CSF is the cancer slope factor for pollutant inhalation (kg·d·mg$^{-1}$). The CSF for long-term inhalation of HCBD recommended by the US EPA were used. IR is the inhalation rate (m$^3$·h$^{-1}$), ET is the exposure time (h·d$^{-1}$), Abs is the absorption fraction of inhaled pollutants, EF is the exposure frequency (d·y$^{-1}$), ED is the exposure time (y), BW is the body weight (kg), and AT is the average time (d). We estimated the increase in cancer risk caused by lifetime exposure to HCBD in different countries and regions of the world using Eq. 9:

$$ADD_i = \frac{C_i \times EF \times ED \times ET}{AT}, \tag{9}$$

where $ADD_i$ is the lifetime average daily exposure (µg·m$^{-3}$).

**Uncertainty**

In this study, HCBD EFs were estimated for 12 different industries. Given the diversity of raw materials, processes, facility operations, and calculation process parameter selection, uncertainty is unavoidable. Consequently, we implemented various measures at different stages to minimize this uncertainty. Regarding sample selection, we collected

samples from a minimum of six plants for each type of industry. The average HCBD concentration was then used to approximate the overall industry concentration. During the sampling process, we carefully selected i-PM samples from operations that had been stable for at least 72 h, which ensured the HCBD concentration was consistent. This meant that our results effectively represented the average emission concentration across each process.

The estimation of i-PM EFs is affected by different standards and processes. In this paper, the EFs for SZn, SPb, SCu, SAl, PCu, IOS, EAF, COP, CK co-processing, and CFPP were derived using the *Emission Source Statistical Survey Emission Accounting Method and Coefficient Manual*[69–76]. These accounting coefficients were established after a statistical survey of industrial sources, and they reflect the typical patterns of pollutant generation and emission for various processes, products, and raw materials under normal operating conditions. The EF for MWI was determined from the latest literature data[77]. The SCu treatment process mainly uses raw materials. Approximately two-thirds of high-grade scrap copper is directly used in the production of copper products without smelting treatment, while one-third of low-grade scrap copper needs smelting treatment. To calculate a comprehensive EF, the average values of high-grade and low-grade scrap copper were used. The primary factor influencing the EF in EAF steelmaking is the type of final product. In this study, carbon steel, which represents 80% of the total output, was selected as a parameter for analysis.

When estimating the HCBD oxidation rate and dry deposition from emission sources, we addressed parameter uncertainties using a worst-case scenario approach. This assumes that $PM_{2.5}$ within the area originates from unintentional emissions from these sources, which allows for calculation of the minimum impact of long-range atmospheric transport by emphasizing local physical or chemical reactions.

## Data availability
All the relevant research data have been included in the Supplementary Information (Supplementary Tables 2, 3, 4).

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

## Acknowledgements

This work was supported by National Natural Science Foundation of China (Grant Numbers 21936007, 22376204, 22076201), the Strategic Priority Research Program of the Chinese Academy of Sciences (Grant Numbers XDB0750400, XDB0750100, XDB0750000), the Second Tibetan Plateau Scientific Expedition and Research Program (STEP) (grant number 2019 QZKK0605). We thank Prof. Haiyan Zhang from Environmental School, Hangzhou Institute for Advanced Study, UCAS for her helpful discussion.

## Author contributions

C.Z. conducted the experimental analysis, analyzed the data and wrote the original draft. L.Y. designed the methodology, analyzed the data and revised the manuscript. A.H. contributed to data discussions. Y.S., C.C., Z.H., Q.Y., and J.Y. conducted the laboratory analysis of samples and the data analysis. M.Z. and G.J. commented on and revised the manuscript. G.L. designed the research, analyzed the data, and wrote and revised the manuscript.

## Competing interests

The authors declare no competing interests.
