## [Peer Review File · Nature Communications]

Atmospheric emissions of hexachlorobutadiene in fine particulate matter from industrial sourcesReviewer #1 (Remarks to the Author):

This is a globally large scale manuscript that investigate the atmospheric emissions of HBCDs from more than 120 full-scale industrial plants in 12 industries. Also, this is the first study reporting the release of unintentional HCBd and source emission factors in global industries. The manuscript is well written and would be of interest to the readership of Nature Communication; however, I believe there are a few areas in which it should be strengthened prior to publication. Specifically, this work would be more impactful if (1) the importance of HBCDs controls was discussed, (2) the novelty of analytical methods in this study was highlighted and (3) the comparison regarding HBCDs concentrations from atmospheric emissions was discussed in the context of previous literature. For these reasons, I recommend acceptance with major revisions. All of the following suggestions are intended to improve the quality of the submitted manuscript.

Major comments:

1. I would highly recommend highlighting the health risk of atmospheric emissions of hexachlorobutadiene earlier in the manuscript, as the readers who are not well familiar with this topic might wonder on multiple occasions while reading through the text.
2. The main concern of this work is lack of the novelty and new insights regarding the analytical methods. I realized that this study was global-scaled; but only the discussion about the concentration of atmospheric emissions of HBCDs from the sampling points is far from sufficient. I would highly suggest the author to strongly strengthen the discussion of their data and provide a more deeply-insightful discussion in the revision.

Reviewer #2 (Remarks to the Author):

Review of "Atmospheric emissions of hexachlorobutadiene in fine particulate matter from industrial sources" by Chenyan Zhao and co-authors.

General comments:

The manuscript "Atmospheric emissions of hexachlorobutadiene in fine particulate matter from industrial sources" by Chenyan Zhao and co-authors investigated the unintentional atmospheric emissions of hexachlorobutadiene (HCBd) in fine particulate matter from 12 industrial sources. We know HCBd is a type of artificial compound and has attracted global attention since it had been listed as controlled persistent organic pollutants (POPs) under the Stockholm Convention in 2015. The identification of HCBd sources is definitely practical significance for environmental protection. The assessment of risk due to industrial HCBd emissions is also helpful towards to sustainable industrial development goals. The authors obtained the unintentional EFs of HCBd for the 12 industries and calculated related emissions on a global scale. The carcinogenic risks of HCBd emissions were also evaluated. This work is meaningful, however there are still some issues that need to be addressed in the current version. My detailed comments are as follows:

Specific comments:

(1) Introduction:

1. The current introduction provides a lot of information but is slightly redundant. More importantly, previous research progress directly related to this study and corresponding limitations lack elaboration, which should be extended. In addition, adding the purpose and significance of this study at the end of the introduction is suggested.
2. Industrial PM was used as a term. How to define the industrial PM? Are the size of PM from industrial sources characterized? What is the size distribution of industrial PM?
3. What types of sources for HCBd emissions were ever emphasized in available references? Why those twelve industrial sources were selected for investigation of HCBd emissions?

(2) Methods:

1. A flowchart to demonstrate the process of this study would be helpful.
2. How do you determine the number of the investigated industrial plants? Are the plant numbers for each source category sufficient for assessment? Where are these 121 sampled full-scale plants located? Specific geographic locations need to be given (perhaps down to the city scale). Are they

distributed globally or just in China? Whether the data collected are sufficiently representative to estimate global emission levels?

3. Lines 376-378: The units of the parameters inside this formula need to be clarified. And the same goes for the rest.

4. Line 379: What are the specific data sources? Please specify.

5. The authors calculated the dry deposition of HCBd with reference to the dry deposition velocity of acenaphthene. The authors state that HCBd is mainly adsorbed on PM_{2.5}, so, why not refer to the dry deposition velocity of PM_{2.5}?

6. Line 414: The sources of the related data for K_i and C_0 should be given.

7. Line 473: Here it is necessary to analyze the effect of different factors on the uncertainty of the results, and it is desirable that the range of uncertainty can be given quantitatively, especially since the data of the present study are all accurate to 0.1g.

(3) Results and discussion

1. Fig. 1. Besides the HCBd concentrations, emission factors of PM, an inclusion of HCBd emission factors is required in Fig. 1. The data of HCBd emission factors should also be provided in the supplement.

2. The investigated industries are mostly thermal related sources. If compared with chemicals manufacturing, what is levels of the thermal related sources?

3. Compared to available inventory of HCBd in China, the emission amount of HCBd from the investigated sources is relatively small. Why did you emphasize the 12 sources in this study?

4. Can the results in this study explain the ultra-high concentrations of HCBd in Mongolia and the Pacific Islands? The sources lead to those ultra-high concentrations might be different from the investigated sources.

5. What is the mechanism of high formation of HCBd in secondary copper smelting, electric arc furnace and hazardous waste incinerations? Or can the reasons leading to high concentrations in those three sources be explained?

6. ¹³C-labelled HCBd was used for internal standard of instrumental analysis. Can a GC/MS spectrum be provided?

7. This section should not involve descriptions of the methods, which should be elaborated in the corresponding section of the Methods.

8. It is recommended that this section be augmented with the practical and policy implications of the findings of this study and an outlook for future research

Reviewer #3 (Remarks to the Author):

This study delved into HCBd emissions from 121 industrial plants operating at full scale, belonging to 12 big industries. The collected industrial samples were rare and valuable. The extensive data in this study could provide excellent data support for HCBd emission factors and inventory establishment. The study proposed prioritizing the control of HCBd emissions in industries such as secondary copper smelting, electric arc furnace steelmaking, and hazardous waste incineration. The results can be beneficial for international convention implementation. This study is well designed and could improve the understanding of HCBd source emissions. Moreover, the discussion about the impact of environmental behaviors such as dry deposition and oxidation reactions on HCBd transportation is interesting and could provide important supporting information for assessing the potential risks of HCBd exposure. Minor revision is recommended before it could be accepted for publication. Detailed suggestions were shown below:

1. The formation mechanisms of HCBd, as well as the influence of factors such as processing temperature and degradation temperature, need to be included and discussed. The different HCBd concentrations in different industrial PM from various sources should be explained from the viewpoint of formation mechanisms and factors.

2. HCBd is a kind of byproducts from chemical manufacturing industries. The emission concentrations and emission factors of HCBd from chemical manufacturing industries should be further added and discussed. HCBd emissions from typical chemical manufacturing processes with the investigated sources should be compared.

3. Particle matter (PM) released from industrial emissions is a crucial factor influencing the occurrence and human exposure of HCBd in the environment. However, the manuscript lacks

detailed descriptions of the characteristics of industrial PM from various industries. Characterizing and clarifying the PM from 12 different industrial sources is essential, as it may affect their transportation behavior in the atmosphere. Furthermore, will the distinct characteristics of i-PM from diverse industrial sources impact the dry deposition behavior?

4. In line 180-182, why compared HCBD EF from coking processes with that from pentachloroene and hexachlorobenzene? Further statement is suggested.

5. It is interesting to evaluate the impact of dry deposition on the transportation and persistence of HCBD. Please explain why dry deposition has no obvious effect on HCBD loss in the atmosphere. Were the consumption pathways (dry deposition, oxidation reactions, long-range atmospheric transport [LART]) of HCBD on i-PM different for different industrial sources (Figure 5)?

6. In Line 253, the authors mentioned that "higher emissions of HCBD in the Global South than in the Global North indicate higher HCBD exposure for individuals in the Global South compared to those in the Global North." The lifetimes of HCBD might be relatively shorter in the Global South compared to the Global North. This should be mentioned when assessing the exposure risks in these regions.

Response to Reviewers' Comments

Reviewer #1:

General Comments:

This is a globally large-scale manuscript that investigate the atmospheric emissions of HCBDs from more than 120 full-scale industrial plants in 12 industries. Also, this is the first study reporting the release of unintentional HCBD and source emission factors in global industries. The manuscript is well written and would be of interest to the readership of Nature Communication; however, I believe there are a few areas in which it should be strengthened prior to publication. Specifically, this work would be more impactful if (1) the importance of HCBDs controls was discussed; (2) the novelty of analytical methods in this study was highlighted; (3) the comparison regarding HCBDs concentrations from atmospheric emissions was discussed in the context of previous literature.

Response: We appreciate the reviewer's very positive comments on the originality and interests of our large-scale research data to the readership of Nature Communication.

Firstly, regarding the importance of HCBD control, we have made supplements and discussion in the revised manuscript (Line109-115, Page5) as shown below.

Hexachlorobutadiene (HCBD) has attracted increasing concerns for its toxicity and health risk under the important international treaties including Stockholm Convention. The production and use of HCBD have been prohibited in most countries worldwide, but its unintentional releases from ongoing global industrial activities will continue along with the human industrial productions. Nevertheless, research on the sources of HCBD emissions remains limited, and the relevant laws and regulations are not fully developed. This study is the first to clarify HCBD unintentional emission inventory globally from industrial sources. It represents the first comprehensive assessment of global unintentional HCBD emissions from industrial sources, elucidating the emission inventory and evaluating potential human health risks associated with these emissions. These results underscore the significance of unintentional HCBD releases from industrial sources in contributing to environmental pollution and human health risks. Consequently, effective control of HCBD emissions from industrial activities is imperative for mitigating human exposure and health risks, as well as for fostering sustainable industrial development.

Moreover, the following aspects that might be helpful for HCBD control was discussed in the

manuscript. (1) The emission levels and emission factors of HCBd from different industries were compared, offering a comprehensive benchmark for recognizing the priority-controlled sources of unintentional HCBd; (2) The formation of HCBd in different process stages for high emission factor industries is investigated, which is helpful for mitigating the unintentional generation of HCBd from the viewpoint of process optimization.

Secondly, regarding the analytical methods, we further highlighted the novelty of analytical methods in the revised manuscript (Line402-418, Page18) as shown below.

We summarized the extraction and purification methods for HCBd determination in solid samples in recent years (**Table R1** as shown below). In order to ensure that the target is extracted completely, each sample was extracted with acetone/n-hexane (1:1, v/v) by Soxhlet extraction for 12 h and then reduced to 1 mL by rotary evaporation. "A multilayer column comprising 2 g of silica gel, 4 g of Florisil, and 4 g of silica gel (from bottom to top) was selected as the optimal clean-up method due to its superior purification efficiency¹. Isotopic dilution GC-MS techniques are the internationally recognized optimum method for trace analysis of persistent organic pollutants (POPs) in complex environmental matrices. Many international analytical standards, such as US EPA method TO-13A and ISO 23646, recommended and used isotopic dilution GC-MS techniques. In our study, for more precise quantification, we also adopted isotopic dilution GC-MS techniques for HCBd quantification. Prior to sample pretreatment, ¹³C-labelled HCBd used as internal standard was spiked into the samples. ¹³C-labelled HCBd standard was utilized for accurate quantitation of HCBd in samples. The HCBd concentration in the sample was determined by gas chromatography mass spectrometry (7890A GC/5975 MSD; Agilent Technologies, Santa Clara, CA) using an HP-5MS column (30 m × 0.25 mm i.d., 0.25 μm film thickness; Agilent Technologies).

Thirdly, regarding the comparison of HCBds concentrations from atmospheric emissions with previous literature, we further supplemented the discussions in the revised manuscript.

(Line187-199, Page9) Currently, there is very limited research on the atmospheric content of hexachlorobutadiene (HCBd), with only a few studies primarily focusing on HCBd concentrations in industrialized regions. The urban atmosphere, characterized by a high concentration of petroleum enterprises², has been found to contain HCBd levels ranging from 4.98 to 20.20 μg/m³. In two studies conducted in 2017 and 2018 near organochlorine pesticide factories, elevated concentrations of HCBd were detected, with levels ranging from 0.03 to 0.33 ng/m³ and from 0.01 to 2.23 ng/m³, respectively. In 2018, HCBd concentrations of 0.21 μg/m³ was detected near a chlor-alkali plant in Catalonia, Spain³.

(Line173-182, Page8) In previous study, the direct quantification of HCBD from industrial emission sources is only available for waste incineration sources. HCBD was detected in fly ash and flue gas samples obtained from industrial waste incineration⁴, exhibiting concentrations of 0.25 ng/g dw and 8.2 ng/Nm³, respectively. The study conducted by Zhang et al. have similar findings, with the concentration of HCBD in municipal solid waste incineration ranging from 1.39 to 7.81 ng/g, while hazardous waste incineration exhibited a higher concentration of 43.8 ng/g. Those results are in the same order of magnitude as our data for waste incineration sources.

However, this is the first monitoring of HCBD releases for other industrial sources. Therefore, there is no available literatures that can be used for the comparison with our results about HCBD emissions from other industrial sources.

Table R1. Methods for HCBD determination in solid samples and its concentrations

(Supplementary Table 5)

Sample	Concentration	Extraction method	Extraction solvent	Clean-up method	Reference
indoor dust		UAE	Hex/acetone	Florisil SPE	3
Fly ash in municipal solid waste	1.39-7.81 ng/g	UAE	DCM	multilayer column: Florisil, silicagel, Na ₂ SO ₄	5
Fly ash in industrial solid waste	43.8 ng/g	UAE	DCM	multilayer column: Florisil, silicagel, Na ₂ SO ₄	5
Fly ash in rotary kilns	0.25 ng/g	Soxhlet extraction	DCM	multilayer column	4
air-dried soil		ASE	DCM/Hex	multilayer column: Florisil, silicagel, Na ₂ SO ₄ , activated copper	6
freeze-dries soil		soxhlet extraction	Hex/acetone	3% water activated Florisil column	7

UAE: ultrasonic-assisted extraction; DCM: dichloromethane; Hex: n-hexane; ASE: accelerated solvent extraction

References:

[1]. Zhang, H. et al. Determination of hexachlorobutadiene, pentachlorobenzene, and hexachlorobenzene in waste incineration fly ash using ultrasonic extraction followed by column cleanup and GC-MS analysis. *Anal. Bioanal. Chem.* 410, 1893–1902 (2018).

[2]. Nan et al. Pollution characters and health risk assessment of VHCs in ambient air in Zhengzhou. *Environmental Pollution & Control* 38, 72–78 (2016).

[3]. van Drooge, B. L., Marco, E. & Grimalt, J. O. Atmospheric pattern of volatile organochlorine compounds and hexachlorobenzene in the surroundings of a chlor-alkali plant. *Sci. Total Environ.* 628–629, 782–790 (2018).

[4]. Kajiwara, N. et al. Environmentally sound destruction of hexachlorobutadiene during waste incineration in commercial- and pilot-scale rotary kilns. *J. Environ. Chem. Eng.* 7, 103464 (2019).

[5]. Zhang, H. et al. Determination of hexachlorobutadiene, pentachlorobenzene, and hexachlorobenzene in waste incineration fly ash using ultrasonic extraction followed by column cleanup and GC-MS analysis. *Anal Bioanal Chem* 410, 1893–1902 (2018).

[6]. Zhang, H. et al. Levels and Distributions of Hexachlorobutadiene and Three Chlorobenzenes in Biosolids from Wastewater Treatment Plants and in Soils within and Surrounding a Chemical Plant in China. *Environ. Sci. Technol.* 48, 1525–1531 (2014).

[7]. Fang, Y. et al. Organochlorine pesticides in soil, air, and vegetation at and around a contaminated site in southwestern China: Concentration, transmission, and risk evaluation. *Chemosphere* 178, 340–349 (2017).

Major comments:

Comment 1: I would highly recommend highlighting the health risk of atmospheric emissions of hexachlorobutadiene earlier in the manuscript, as the readers who are not well familiar with this topic might wonder on multiple occasions while reading through the text.

Response: Thanks for your comment. We have supplemented and highlighted the health risk of HCBD in the revised manuscript (**Line 43-46, Page 2**) according to the reviewer' comments, which were also shown below.

“Since 1978, HCBD has been recognized as the most hazardous aliphatic halogenated hydrocarbon, capable of inducing epithelial necrotizing nephritis ^[1]. The occurrences of HCBD in the atmosphere may potentially lead to respiratory damage and carcinogenic effects. Understanding the source emissions of HCBD into atmosphere is pivotal for recognizing their respiratory exposures and potential health risks.”

Reference:

[1] Duprat, P. & Gradiski, D. Percutaneous Toxicity of Hexachlorobutadiene. *Acta Pharmacologica Et Toxicologica* 43, 346–353 (1978).

Comment 2: The main concern of this work is lack of the novelty and new insights regarding the analytical methods. I realized that this study was global-scaled; but only the discussion about the concentration of atmospheric emissions of HBCDs from the sampling points is far from sufficient. I would highly suggest the author to strongly strengthen the discussion of their data and provide a more deeply-insightful discussion in the revision.

Response: Thanks for your comments. The novelty of analytical methods for HCBd in this study embodies in the application of ^{13}C -labelled internal standard, which ensure accurate qualification and quantification of HCBd in the industrial PM. In the previous limited studies on HCBd analysis, which were summarized in **Table R1**, HCBd analysis were mainly conducted by external standard method. In the external standard method, the samples and the standard solutions are measured separately. This leads to the inevitable accumulation of errors in sample preparation and measurement, thereby compromising the accuracy and precision of the results. We referred to the internationally recognized optimum method for trace analysis of persistent organic pollutants (POPs) in complex environmental matrices, such as US EPA method TO-13A and ISO 23646. We adopted isotopic dilution GC-MS techniques for HCBd quantification. Prior to sample pretreatment, ^{13}C -labelled HCBd used as internal standard was spiked into the samples. By comparing the ratios of the analyte and isotope signals, the actual concentration of HCBd can be accurately determined, even in the presence of matrix effects or other interference factors. Relevant descriptions on the analytical methods has been added in the revised manuscript (*Line402-418, Page17-18*).

“In order to ensure that HCBd is extracted completely, each sample was extracted with acetone/n-hexane (1:1, v/v) by Soxhlet extraction for 12 h and then reduced to 1 mL by rotary evaporation. A multilayer column comprising 2 g of silica gel, 4 g of Florisil, and 4 g of silica gel (from bottom to top) was selected as the optimal clean-up method due to its superior purification efficiency¹. Isotopic dilution GC-MS techniques are the internationally recognized optimum method for trace analysis of persistent organic pollutants (POPs) in complex environmental matrices. Many international analytical standards, such as US EPA method TO-13A and ISO 23646, recommended and used isotopic dilution GC-MS techniques. In our study, for more precise quantification, we also adopted isotopic dilution GC-MS techniques for HCBd quantification. Prior to sample pretreatment, ^{13}C -labelled HCBd used as internal standard was spiked into the samples. ^{13}C -labelled HCBd standard was utilized for accurate quantitation of HCBd in samples. The HCBd concentration in the sample was determined by gas chromatography mass spectrometry (7890A GC/5975 MSD; Agilent Technologies, Santa Clara, CA) using an HP-5MS

column (30 m × 0.25 mm i.d., 0.25 µm film thickness; Agilent Technologies).”

In addition, this study is the first to clarify HCBD unintentional emission inventory globally from industrial sources. The global-scaled emissions of HBCD emissions were present in Fig. 3 in the revised manuscript. According to the reviewer’s comments, we have further strengthened the discussion of our data in the revised manuscript (*Line231-235, Page 10; Line262-273, Page 11-12*). The added discussion about the HCBD emissions from industrial activities were provided below.

“Figure 2 shows HCBD emissions from different countries across six controlled industries. Among the emission sources, SCu was the largest with emission of approximately 2643.3 g of HCBD. This was attributed to the extremely high concentration of HCBD in i-PM from this industry. China, the USA, and Japan are the main contributors to HCBD atmospheric emissions from SCu.”

“Figure 3 shows the global atmospheric emissions of HCBD in 2018. A total of 101 countries and territories emitted 8452.8 g of HCBD in industrial i-PM. A total of 101 countries and territories emitted 8452.8 g of HCBD in industrial i-PM. With the continuous growth of the global population, urbanization is accelerating, leading to a gradual increase in population density. Simultaneously, as population density rises, so does industrial demand. We link HCBD emissions with population data and calculate per capita emissions of HCBD in different countries. Higher per capita emissions are primarily observed in countries with higher GDP and lower population density, such as Belgium (9.8 µg/per), Australia (8.8 µg/per), Sweden (4.2 µg/per), and South Korea (3.3 µg/per). The emissions of POPs are significantly influenced by the industrial capacity of different countries. The level of industrial production, particularly in developed nations, is closely associated with economic development. Therefore, economic indicators, such as GDP, can exert a substantial impact on pollutant emissions.”

Reviewer #2:

Review of “Atmospheric emissions of hexachlorobutadiene in fine particulate matter from industrial sources” by Chenyan Zhao and co-authors.

General comments:

The manuscript “Atmospheric emissions of hexachlorobutadiene in fine particulate matter from industrial sources” by Chenyan Zhao and co-authors investigated the unintentional atmospheric emissions of

hexachlorobutadiene (HCBD) in fine particulate matter from 12 industrial sources. We know HCBD is a type of artificial compound and has attracted global attention since it had been listed as controlled persistent organic pollutants (POPs) under the Stockholm Convention in 2015. The identification of HCBD sources is definitely practical significance for environmental protection. The assessment of risk due to industrial HCBD emissions is also helpful towards to sustainable industrial development goals. The authors obtained the unintentional EFs of HCBD for the 12 industries and calculated related emissions on a global scale. The carcinogenic risks of HCBD emissions were also evaluated. This work is meaningful, however there are still some issues that need to be addressed in the current version. My detailed comments are as follows:

Response: We sincerely appreciate the reviewer's very positive comments on the practical significance of this study for environmental protection and sustainable industrial development goals. We have carefully revised our manuscript according to the reviewer's comments. Detailed responses to the comments are provided below.

Comment (1). Introduction:

Specific comment 1. The current introduction provides a lot of information but is slightly redundant. More importantly, previous research progress directly related to this study and corresponding limitations lack elaboration, which should be extended. In addition, adding the purpose and significance of this study at the end of the introduction is suggested.

Response: Thanks for the reviewer's comments. According to the reviewer's comments, we have simplified the Introduction part and focused on the current progress of HCBD occurrence in two dominant HCBD sources, including commercial chemical manufacturing processes and industrial activities. Limitations of current knowledge of HCBD has been supplemented and elaborated. The biggest limitation is that the field studies of HCBD emissions from industrial sources were almost gap, inducing the lacking HCBD emission data from industrial sources. Relevant description of the current progress and limitations of HCBD were shown in the revised Introduction part (**Line 74-75, Page 5; Line 106-108, Page 5**), which were also shown below.

"However, current HCBD emission data from industrial sources were very limited."

"This study is the first to clarify HCBD unintentional emission inventory globally from industrial sources and will be helpful for formulating effective international strategies of HCBD control."

This study is the first to derive HCBd emission factors for 12 industrial sources, which could contribute to compilation of a global inventory. The spatial distribution of HCBd emissions on a national scale were mapped and the potential carcinogenic risk of HCBd globally were estimated. This study can fill the gap of unintentionally emissions of HCBd from industrial sources and will be helpful for formulating effective international strategies of HCBd control. Relevant description on the purpose and significance of this study have been added at the end of the introduction in the revised manuscript (Line 65-79, Page 4; Page 99-101, Page 5), which were also shown below in blue text.

“Currently, a few studies have reported the release of HCBd from chemical production processes that use chlorine and waste disposal ^[1,2]. Our previous study has also investigated the occurrence of HCBd in products and bottom liquid of chlorobenzene, trichloroethylene, and tetrachloroethylene chemical manufacturing plants ^[2]. HCBd concentrations in the bottom liquid samples contributed 24%–99% of the total HCBd formed in the chemical production plants. The bottom liquid was disposed of as hazardous waste by incineration ^[2]. Therefore, a proportion of HCBd from commercial chemical manufacturing processes would finally enter into environment by the unintentional releases from incinerations of bottom liquid. However, current HCBd emission data from industrial sources were very limited. It has also been found that HCBd occurred in industrial fine particulate matter (i-PM) from waste incineration facilities in East China ^[3].”

“The EFs for the 12 industries were derived according to the UNEP methodology, which will be essential data for compiling a global emission inventory.”

References:

[1] Wang, L., Bie, P. & Zhang, J. Estimates of unintentional production and emission of hexachlorobutadiene from 1992 to 2016 in China. *Environmental Pollution* **238**, 204–212 (2018).

[2] Wang, M. et al. Hexachlorobutadiene emissions from typical chemical plants. *Front. Env. Sci. Eng.* **15**, 60 (2021).

[3] Zhang, H. et al. Determination of hexachlorobutadiene, pentachlorobenzene, and hexachlorobenzene in waste incineration fly ash using ultrasonic extraction followed by column cleanup and GC-MS analysis. *Anal Bioanal Chem* **410**, 1893–1902 (2018).

Specific comment 2. Industrial PM was used as a term. How to define the industrial PM? Are the size of PM from industrial sources characterized? What is the size distribution of industrial PM?

Response: Thanks for your comment. The samples in our study were obtained from industrial particulate matters produced during high-temperature processes of 12 different industrial sources. In our previous

study, we used scanning electron microscope (SEM) to characterize the fundamental properties of these industrial particles, which primarily ranged in size less than 2.5 μm ($\text{PM}_{2.5}$). $\text{PM}_{2.5}$ released by most of the industrial sources account for 90.1–100% of the total PM emissions, with an average of 97.9% and median of 99.5%. Thus, those particles can be considered as industrial fine PM. The SEM results have been provided in the materials to *Nature Communications* (DOI: 10.1038/s41467-023-39491-5).

Figure R1. The SEM photography of collected industrial PM.

Specific comment 3. What types of sources for HCBD emissions were ever emphasized in available

references? Why those twelve industrial sources were selected for investigation of HCBd emissions?

Response: Thanks for your comment. HCBd has no known natural sources. Production and use of HCBd as chemical have been stopped because of its ecotoxicity. Current emissions of HCBd primarily occur as unintentional releases during industrial activities.

In the past, many studies have confirmed that the typical source of HCBd is as a by-product in the production of chlorinated chemicals including chlorobenzenes, trichloroethylene and tetrachloroethylene^{8-10,3,6}, but few studies of HCBd formation during chemical production processes have been performed. In our previous study, we detected HCBd concentrations in three mixed chlorinated chemical plants at different process stages¹¹ (in raw materials, intermediate products, final products, and bottom residues). The highest concentration of HCBd was found in the bottom residues of factories producing trichloroethylene and tetrachloroethylene, reaching 243,000 mg/ml. In contrast, the HCBd content in the final products, trichloroethylene and tetrachloroethylene samples were lower at 0.076 and 0.071 mg/ml, respectively. These residue with high content of HCBd will be incinerated as hazardous waste.

Unintentional emission is another significant source of HCBd. In the "Risk Management Evaluation on Hexachlorobutadiene" released by UNEP, HCBd can be unintentionally produced during combustion and other thermal and industrial processes. According to the information of U.S. Toxic Substances Emission Inventory (TRI) in 2011, cement manufacturing is expected to emit a certain amount of HCBd into the atmosphere, constituting 10.73% of the total emissions. However, its mechanism and the influencing factors on emissions are yet to be studied.

The research on the emission sources of HCBd is relatively old and most of them are before the implementation of the Stockholm Convention. Therefore, the identification and quantification of new industrial sources of HCBd is emerging required. According to the 'Identification and Quantification Toolkit for the Release of Dioxins, Furans, and Other Unintentional POPs,' released by the United Nations Environment Programme and the Stockholm Convention, twelve potential important thermal industrial sources were selected and investigated in our research. These sources essentially encompass most industrial sources that could offer favorable conditions for HCBd generation, and we detected the concentration levels of HCBd in the PM emitted by these industrial sources.

Moreover, most of chemical plants release HCBd in gas-phase or as impurity in chemical productions, while industrial sources investigated in this study release HCBd absorbed in the particle matters. Therefore, it is important and essential to understanding the HCBd inventory in industrial PM

for better understanding their exposure.

Comment (2) Methods:

Specific comment 1. A flowchart to demonstrate the process of this study would be helpful.

Response: we have supplemented the flow chart of this study in Supplementary Information. (SI).

Figure R2. The flowchart of this study

(Supplementary Figure 2)

Specific comment 2. How do you determine the number of the investigated industrial plants? Are the plant numbers for each source category sufficient for assessment? Where are these 121 sampled full-scale plants located? Specific geographic locations need to be given (perhaps down to the city scale). Are they distributed globally or just in China? Whether the data collected are sufficiently representative to estimate global emission levels?

Response: Thanks for your comment. To gain a more comprehensive assessment, we have made our greatest efforts to collect as many as samples from industrial plants. At least 6 plants were investigated for each source category and the total sampled plants reached to 121. To our best know, the data scale in this study is considerable large compared to other industrial studies. Even in *Toolkit for Identification and Quantification of Releases of Dioxins, Furans and Other Unintentional POPs* issued by UNEP and Stockholm Convention, emission factor is the average of the emission factors from only several plants. The plant numbers for each source category in this study is sufficient for assessment.

All the samples were collected from plants in China. The geographic locations of investigated industrial plants were shown below in the **Figure R3**. However, considering the commonality of global industrial production technologies and China's major production status in these industries, results of HCBd in those samples collected from 121 industrial plants in China are considered to be still representative enough to estimate global emission levels.

Figure R3. The scheme of geographic locations of investigated industrial plants (**Supplementary Figure 3**)

Specific comment 3. Lines 376-378: The units of the parameters inside this formula need to be clarified. And the same goes for the rest.

Response: Thanks for your comment. We have made clarification and supplements according to the reviewer's comments.

Line 433: $E = \bar{C}_i \times EF_i \times Y_i,$ (1)

where E is the annual emission of HCBD in each industry (ng); \bar{C}_i is the average value of the HCBD concentration for each plant in the same industry (ng/g); and EF_i (kg/t) and Y_i (kt) are the average EF of i-PM and annual production yield in the specific industry, respectively. The Y_i data were collected from relevant international statistics organization. We multiplied the industrial activity levels by the fraction of each emission sector to obtain the emission inventory for each emission sector.

$$D_{p,i} = m \times Y_i = \frac{Dp}{Y_0} \times Y_i, \quad (4)$$

where Y_i (kt) is the total output of industry i, and Y_0 (kt) is the annual output of the individual plant.”

Specific comment 4. Line 379: What are the specific data sources? Please specify.

Response: Thanks for your comment. We have provided the sources for specific data in Supplementary Information as shown below.

Table R2. Annual production yield data sources for 12 different industries
(Supplementary Table 7)

	Website Name/Institution name	URL
SZn	U.S. Geological Survey.	https://www.usgs.gov/
SPb	U.S. Geological Survey.	https://www.usgs.gov/
SCu	International Copper Study Group	https://icsg.org/
PCu	International Copper Study Group	https://icsg.org/
SAI	International Aluminium institute.	https:// alucycle.world-aluminium.org/
HWI	The Word Bank.	https:// datacatalog.worldbank.org/
MWI	The Word Bank.	https:// datacatalog.worldbank.org/
EAF	World Steel Association.	https://www.worldsteel.org/
COP	The Word Bank.	https:// datacatalog.worldbank.org/
CCK	U.S. Geological Survey.	https://www.usgs.gov/
CFPP	Energy Institute	https://www.energyinst.org/
IOS	World Steel Association.	https://www.worldsteel.org/

Specific comment 5. The authors calculated the dry deposition of HCBD with reference to the dry deposition velocity of acenaphthene. The authors state that HCBD is mainly adsorbed on PM_{2.5}, so, why not refer to the dry deposition velocity of PM_{2.5}?

Response: Thank you for your comment. In this study, acenaphthene with similar boiling point to HCBD was used to estimate the dry deposition near the factory. This choice was primarily motivated by the fact that although HCBD is released into the atmosphere in adsorbed particle form, gas-solid exchange processes can occur within complex atmospheric environments. Therefore, we opted to utilize the sedimentation rate of HCBD under actual environmental conditions to mitigate potential interference from complex environmental factors. This approach is closer to HCBD's sedimentation behavior in real-

environmental settings and enables a more accurate quantification of its contribution to both transport and persistence near emission sources.

Specific comment 6. Line 414: The sources of the related data for K_i and C_0 should be given.

Response: Thank you for your comment. The relevant data used in this study are derived from "*The experimental observation, mechanism, and kinetic studies on the reaction of hexachloro-1,3-butadiene initiated by typical atmospheric oxidants*" published in the *Science of the Total Environment*^[1]. We have provided the data in **Table R3** as shown below.

Table R3. Reaction rate constants of different oxidants with HCBD and their contribution to the total oxidation amount

(Table 2 in the manuscript)

Oxidants	OH	Cl	NO ₃	HO ₂
K_i (cm ³ ·mol ⁻¹ ·s ⁻¹) [1]	6.33×10 ⁻¹⁶	4.51×10 ⁻¹³	1.32×10 ⁻²⁰	4.33×10 ⁻²⁹
C_0 (mol·cm ⁻³) [1]	1.4×10 ⁶	9.0×10 ³	4.81×10 ⁷	1.2×10 ⁸
Contribution (%)	17.92	82.08	1.07×10 ⁻³	6.59×10 ⁻⁶

Reference:

[1] Zhang, X., Yang, M., Sun, X., Wang, X. & Wang, Y. *The experimental observation, mechanism and kinetic studies on the reaction of hexachloro-1,3-butadiene initiated by typical atmospheric oxidants. Science of The Total Environment* 2018, **627**, 256–263.

Specific comment 7. Line 473: Here it is necessary to analyze the effect of different factors on the uncertainty of the results, and it is desirable that the range of uncertainty can be given quantitatively, especially since the data of the present study are all accurate to 0.1 g.

Response: We sincerely appreciate the reviewer's suggestions. In the study, we used the emission factor method to evaluate HCBD emissions from the investigated industries. As the reviewer commented, different factors affect the degree of uncertainty in the results. Firstly, the emissions survey is based on Chinese factories, which might be different in the management with other regions. However, in terms of production scale, Chinese factories have very large output (or combustion volume), which can represent the general level of the world. These accounting coefficients were established after a statistical survey of

industrial sources, and they reflect the typical patterns of pollutant generation and emission for various processes, products, and raw materials under normal operating conditions. China is a large industrial country, and there are many regulations that limit the emission of particulate matter at the end-of-pipe of industrial sources. Moreover, considering the commonality of global industrial production technologies and China's major production status in these industries, results of HCBD in those samples collected from 121 industrial plants in China are considered to be still representative to estimate global emission levels.

Quantitative uncertainty requires actual data from around the world, but emissions are affected by many factors, and a rash quantitative description may be misleading, and more research is needed to determine. However, only in terms of concentration, we improved the accuracy in order to better understand the data and make a preliminary assessment of HCBD global emissions. Field study on full-scale industrial plants normally require complex negotiations with the industrial operators. Moreover, there are relatively more variables for full-scale industrial plants. In this study, we conducted the evaluation according to the inventory methodology recommended by the European Monitoring and Evaluation Programme/European Environment Agency (EMEP/EEA)^[1] and United Nations Environment Programme (UNEP)^[2]. We sincerely hope the reviewer understand the difficulties in conducting field study on full-scale industrial plants. Thanks again for the reviewer's comments.

References:

[1] EMEP. EMEP/EEA air pollutant emission inventory guidebook 2019. <https://www.eea.europa.eu/publications/emep-eea-guidebook-2019/download>.(2019)

[2] UNEP. Toolkit for Identification and Quantification of Releases of Dioxins, Furans and Other Unintentional POPs under Article 5 of the Stockholm Convention. <https://www.pops.int/Implementation/UnintentionalPOPs/ToolkitforUPOPs/Overview/tabid/372/Default.aspx>(2013).

Comment (3) Results and discussion

Specific comment 1. Fig. 1. Besides the HCBD concentrations, emission factors of PM, an inclusion of HCBD emission factors is required in Fig. 1. The data of HCBD emission factors should also be provided in the supplementary information.

Response: Thank you for your comment. We have made additions (the right part) to **Figure 1** in the revised manuscript according to the reviewer's suggestion. The data of HCBD emission factors were also provided in the **Supplementary Table 2**. The relevant modifications are also shown below.

Figure R4. The revised Figure 1 in the revised manuscript.

Table R4 Detail information on collected samples in this study

(Supplementary Table 2)

Number	Sample ID	Concentration(ng/g)	Industrial activity	Raw material	Sample style	Average emission factors of HCBDD (mg/kt • production)
1	CFPP1	3.86	Coal-Fired Power Plant	Bituminous coal, anthracite and lignite	Particulate matter	0.05
2	CFPP2	2.13	Coal-Fired Power Plant	Bituminous coal, anthracite and lignite	Particulate matter	
3	CFPP3	1.47	Coal-Fired Power Plant	Bituminous coal, anthracite and lignite	Particulate matter	
4	CFPP4	7.48	Coal-Fired Power Plant	Bituminous coal, anthracite and lignite	Particulate matter	
5	CFPP5	n.d.	Coal-Fired Power Plant	Bituminous coal, anthracite and lignite	Particulate matter	
6	CFPP6	6.14	Coal-Fired Power Plant	Bituminous coal, anthracite and lignite	Particulate matter	
7	CFPP7	0.14	Coal-Fired Power Plant	Bituminous coal, anthracite and lignite	Particulate matter	
8	CFPP8	1.81	Coal-Fired Power Plant	Bituminous coal, anthracite and lignite	Particulate matter	
9	CFPP9	n.d.	Coal-Fired Power Plant	Bituminous coal, anthracite and lignite	Particulate matter	
10	CFPP10	11.92	Coal-Fired Power Plant	Bituminous coal, anthracite and lignite	Particulate matter	
11	CFPP11	1.42	Coal-Fired Power Plant	Bituminous coal, anthracite and lignite	Particulate matter	
12	CFPP12	7.48	Coal-Fired Power Plant	Bituminous coal, anthracite and lignite	Particulate matter	
13	CFPP13	46.25	Coal-Fired Power Plant	Bituminous coal, anthracite and lignite	Particulate matter	
14	CFPP14	3.05	Coal-Fired Power Plant	Bituminous coal, anthracite and lignite	Particulate matter	
15	CFPP15	6.71	Coal-Fired Power Plant	Bituminous coal, anthracite and lignite	Particulate matter	

						0.14
16	CK1	0.60	co-processing in cement kilns	Hazardous Waste	Particulate matter	
17	CK2	1.03	co-processing in cement kilns	Hazardous Waste	Particulate matter	
18	CK3	2.23	co-processing in cement kilns	Hazardous Waste	Particulate matter	
19	CK4	1.06	co-processing in cement kilns	Hazardous Waste	Particulate matter	
20	CK5	9.96	co-processing in cement kilns	Hazardous Waste	Particulate matter	
21	CK6	2.94	co-processing in cement kilns	Hazardous Waste	Particulate matter	
22	COP1	3.45	coke production	Coke	Particulate matter	2.46
23	COP2	14.28	coke production	Coke	Particulate matter	
24	COP3	19.03	coke production	Coke	Particulate matter	
25	COP4	13.14	coke production	Coke	Particulate matter	
26	COP5	n.d.	coke production	Coke	Particulate matter	
27	COP6	2.63	coke production	Coke	Particulate matter	
28	COP7	1.32	coke production	Coke	Particulate matter	
29	COP8	n.d.	coke production	Coke	Particulate matter	
30	COP9	0.62	coke production	Coke	Particulate matter	
31	COP10	13.12	coke production	Coke	Particulate matter	
32	COP11	2.49	coke production	Coke	Particulate matter	
33	COP12	1.63	coke production	Coke	Particulate matter	
34	COP13	0.56	coke production	Coke	Particulate matter	
35	EAF1	27.79	Electric Arc Furnace Steelmaking	Scrap	Particulate matter	3.91
36	EAF2	441.55	Electric Arc Furnace Steelmaking	Scrap	Particulate matter	
37	EAF3	18.01	Electric Arc Furnace Steelmaking	Scrap	Particulate matter	
38	EAF4	0.31	Electric Arc Furnace Steelmaking	Scrap	Particulate matter	
39	EAF5	1.49	Electric Arc Furnace Steelmaking	Scrap	Particulate matter	

40	EAF6	0.91	Electric Arc Furnace Steelmaking	Scrap	Particulate matter
41	EAF7	11.14	Electric Arc Furnace Steelmaking	Scrap	Particulate matter
42	EAF8	1.13	Electric Arc Furnace Steelmaking	Scrap	Particulate matter
43	EAF9	n.d.	Electric Arc Furnace Steelmaking	Scrap	Particulate matter
44	EAF10	n.d.	Electric Arc Furnace Steelmaking	Scrap	Particulate matter
45	EAF11	12.16	Electric Arc Furnace Steelmaking	Scrap	Particulate matter
46	EAF12	5.37	Electric Arc Furnace Steelmaking	Scrap	Particulate matter
47	EAF13	0.19	Electric Arc Furnace Steelmaking	Scrap	Particulate matter
48	EAF14	0.70	Electric Arc Furnace Steelmaking	Scrap	Particulate matter
49	HWI1	0.54	hazardous waste incinerator	Hazardous waste such as medical waste	Particulate matter
50	HWI2	1.07	hazardous waste incinerator	Hazardous waste such as medical waste	Particulate matter
51	HWI3	0.27	hazardous waste incinerator	Hazardous waste such as medical waste	Particulate matter
52	HWI4	39.93	hazardous waste incinerator	Hazardous waste such as medical waste	Particulate matter
53	HWI5	0.21	hazardous waste incinerator	Hazardous waste such as medical waste	Particulate matter
54	HWI6	47.40	hazardous waste incinerator	Hazardous waste such as medical waste	Particulate matter
55	HWI7	1.30	hazardous waste incinerator	Hazardous waste such as medical waste	Particulate matter
56	HWI8	0.73	hazardous waste incinerator	Hazardous waste such as medical waste	Particulate matter
57	HWI9	0.95	hazardous waste incinerator	Hazardous waste such as medical waste	Particulate matter
58	HWI10	0.10	hazardous waste incinerator	Hazardous waste such as medical waste	Particulate matter

0.23

59	HW111	176.56	hazardous waste incinerator	Hazardous waste such as medical waste	Particulate matter	
60	HW112	27.05	hazardous waste incinerator	Hazardous waste such as medical waste	Particulate matter	
61	HW113	0.65	hazardous waste incinerator	Hazardous waste such as medical waste	Particulate matter	
62	HW114	28.83	hazardous waste incinerator	Hazardous waste such as medical waste	Particulate matter	0.42
63	IOS1	n.d.	iron ore sintering	Iron Ore	Particulate matter	
64	IOS2	n.d.	iron ore sintering	Iron Ore	Particulate matter	
65	IOS3	n.d.	iron ore sintering	Iron Ore	Particulate matter	
66	IOS4	17.26	iron ore sintering	Iron Ore	Particulate matter	
67	IOS6	1.88	iron ore sintering	Iron Ore	Particulate matter	
68	IOS7	13.67	iron ore sintering	Iron Ore	Particulate matter	
69	IOS8	12.59	iron ore sintering	Iron Ore	Particulate matter	
70	MW11	0.00	municipal solid waste incineration	Municipal solid waste	Particulate matter	0.03
71	MW12	0.82	municipal solid waste incineration	Municipal solid waste	Particulate matter	
72	MW13	0.95	municipal solid waste incineration	Municipal solid waste	Particulate matter	
73	MW14	10.62	municipal solid waste incineration	Municipal solid waste	Particulate matter	
74	MW15	1.32	municipal solid waste incineration	Municipal solid waste	Particulate matter	
75	MW16	0.53	municipal solid waste incineration	Municipal solid waste	Particulate matter	
76	MW17	n.d.	municipal solid waste incineration	Municipal solid waste	Particulate matter	
77	MW18	12.46	municipal solid waste incineration	Municipal solid waste	Particulate matter	
78	MW19	n.d.	municipal solid waste incineration	Municipal solid waste	Particulate matter	
79	MW110	n.d.	municipal solid waste incineration	Municipal solid waste	Particulate matter	

80	MW111	0.36	municipal solid waste incineration	Municipal solid waste	Particulate matter	
81	PCu1	n.d.	Primary Copper Smelting	copper ore	Particulate matter	1.30
82	PCu2	9.40	Primary Copper Smelting	copper ore	Particulate matter	
83	PCu3	1.95	Primary Copper Smelting	copper ore	Particulate matter	
84	PCu4	8.59	Primary Copper Smelting	copper ore	Particulate matter	
85	PCu5	n.d.	Primary Copper Smelting	copper ore	Particulate matter	
86	PCu6	8.26	Primary Copper Smelting	copper ore	Particulate matter	
87	SA11	n.d.	Secondary Aluminum Smelting	Waste Aluminum	Particulate matter	2.45
88	SA12	1.25	Secondary Aluminum Smelting	Waste Aluminum	Particulate matter	
89	SA13	6.48	Secondary Aluminum Smelting	Waste Aluminum	Particulate matter	
90	SA14	4.33	Secondary Aluminum Smelting	Waste Aluminum	Particulate matter	
91	SA15	16.89	Secondary Aluminum Smelting	Waste Aluminum	Particulate matter	
92	SA16	1.84	Secondary Aluminum Smelting	Waste Aluminum	Particulate matter	
93	SCu1	20.40	Secondary Copper Smelting	Waste Mixed Copper	Particulate matter	660.85
94	SCu2	2.05	Secondary Copper Smelting	Waste Mixed Copper	Particulate matter	
95	SCu3	18.40	Secondary Copper Smelting	Waste Mixed Copper	Particulate matter	
96	SCu4	3.75	Secondary Copper Smelting	Waste Mixed Copper	Particulate matter	
97	SCu5	10144.37	Secondary Copper Smelting	Waste Mixed Copper	Particulate matter	
98	SCu6	3457.63	Secondary Copper Smelting	Waste Mixed Copper	Particulate matter	

99	SCu7	73.82	Secondary Copper Smelting	Waste Mixed Copper	Particulate matter
100	SCu8	33.49	Secondary Copper Smelting	Waste Mixed Copper	Particulate matter
101	SCu9	36.18	Secondary Copper Smelting	Waste Mixed Copper	Particulate matter
102	SCu10	6.35	Secondary Copper Smelting	Waste Mixed Copper	Particulate matter
103	SPb1	0.35	Secondary Lead Smelting	Waste Lead	Particulate matter
104	SPb2	n.d.	Secondary Lead Smelting	Waste Lead	Particulate matter
105	SPb3	0.33	Secondary Lead Smelting	Waste Lead	Particulate matter
106	SPb4	17.52	Secondary Lead Smelting	Waste Lead	Particulate matter
107	SPb5	1.29	Secondary Lead Smelting	Waste Lead	Particulate matter
108	SPb6	16.14	Secondary Lead Smelting	Waste Lead	Particulate matter
109	SPb7	0.48	Secondary Lead Smelting	Waste Lead	Particulate matter
110	SPb8	0.69	Secondary Lead Smelting	Waste Lead	Particulate matter
111	SPb9	11.09	Secondary Lead Smelting	Waste Lead	Particulate matter
112	SPb10	2.37	Secondary Lead Smelting	Waste Lead	Particulate matter
113	SPb11	1.86	Secondary Lead Smelting	Waste Lead	Particulate matter
114	SZn1	5.49	Secondary Zinc Smelting	Waste Mixed Zinc	Particulate matter
115	SZn2	2.08	Secondary Zinc Smelting	Waste Mixed Zinc	Particulate matter
116	SZn3	0.80	Secondary Zinc Smelting	Waste Mixed Zinc	Particulate matter
117	SZn4	1.68	Secondary Zinc Smelting	Waste Mixed Zinc	Particulate matter

3.22

1.05

118	SZn5	3.05	Secondary Zinc Smelting	Waste Mixed Zinc	Particulate matter
119	SZn6	3.57	Secondary Zinc Smelting	Waste Mixed Zinc	Particulate matter
120	SZn7	n.d.	Secondary Zinc Smelting	Waste Mixed Zinc	Particulate matter
121	SZn8	n.d.	Secondary Zinc Smelting	Waste Mixed Zinc	Particulate matter

Specific comment 2. The investigated industries are mostly thermal related sources. If compared with chemicals manufacturing, what is levels of the thermal related sources?

Response: Thanks for your comment. The chemical manufacturing plants are significant contributors to the formation of HCBD. HCBD can be generated as by-product during the production of chemicals including chlorobenzenes, trichloroethylene and tetrachloroethylene ^[1]. However, there is little information about the emission factors of HCBD in the chemical industry. Previous studies have used HCBD concentration in primary products to represent the emission factor of HCBD, and we summarized the HCBD generation level reported in previous literatures as shown in **Table R4** below. The results demonstrate a reduction in the proportion of HCBD generated in products as a result of process improvement. Furthermore, within the chemical industry, HCBD by-product is not directly discharged into the environment. Studies have shown that 40% will be incinerated as hazardous waste, 20% will be fractionated and recovered, and only a small fraction will be directly discharged into the environment. Therefore, it is difficult to compare the actual emission factors of the chemical industry with the HCBD emission factors in the industrial thermal process. But regarding the comparison of HCBD concentration, the HCBD levels in bottom liquid from trichloroethylene and tetrachloroethylene related industries were obviously higher than that from industrial thermal sources investigated in this study ^[1].

Table R4. HCBD concentrations in the raw product from different chemical industries

(Supplementary Table 1)

Items	Process	Concentration (ng/mL)	Reference	Time
CTC	Methane method	5.00×10^{-2}	UNEP, 2013	1991 ^[3]
	Methane method (optimized)	5.00×10^{-3}	UNEP, 2013	1991 ^[3]
	Methanol method	8.17×10^{-5}	Zhang, 2015	2015 ^[2]
	Methanol method	3.96×10^{-7}	Wang, 2021	2021 ^[1]
HCCP	Cyclopentadiene method	1.11×10^{-2}	UNEP, 2013	1991 ^[3]
PCE	Acetylene method	4.00×10^{-3}	Beijing Normal University, 2014	2014
	CTC method	4.20×10^{-3}	Beijing Normal University, 2014	2014
	Acetylene method	7.60×10^{-8}	Wang, 2021	2021 ^[1]

	CTC method	3.96×10^{-4}	Wang, 2021	2021 ^[1]
TCE	Acetylene method	4.00×10^{-3}	UNEP, 2013	1991 ^[3]
	Acetylene method	7.60×10^{-8}	Wang, 2021	2021 ^[1]

References:

[1] Wang, M. et al. Hexachlorobutadiene emissions from typical chemical plants. *Front. Env. Sci. Eng.* **15**, 60 (2021).

[2]Zhang, L., Yang, W., Zhang, L. & Li, X. Highly chlorinated unintentionally produced persistent organic pollutants generated during the methanol-based production of chlorinated methanes: A case study in China. *Chemosphere* **133**, 1–5 (2015).

[3] UNEP. Risk management evaluation on hexachlorobutadiene. <https://pops.int> (2013).

Specific comment 3. Compared to available inventory of HCBd in China, the emission amount of HCBd from the investigated sources is relatively small. Why did you emphasize the 12 sources in this study?

Response: Thanks for the reviewer’s comments. Current available inventory of unintentional HCBd in China mainly focused on HCBd as by-products during chemical manufacturing processes. Our previous field study on HCBd occurrences in raw materials, intermediate products, products, and bottom residues from chemical manufacturing plants have also found that HCBd concentrations in the bottom liquid samples contributed 24–99% of the total HCBd formed in the chemical manufacturing processes^[1]. Then the bottom liquid was disposed of as hazardous waste by incineration. Therefore, a proportion of HCBd from commercial chemical manufacturing processes would finally enter into environment by the unintentional releases from incinerations of bottom liquid. In addition, other thermal-related industrial sources such as secondary metal smelting processes, iron ore sintering processes, et al., can also unintentionally produce HCBd because of the abundant precursors in raw materials and suitable temperature for HCBd formation. Exploring HCBd unintentional emissions from industrial sources are important for establishing complete HCBd inventory.

In this study, we derived the emission factors of HCBd from the 12 industrial sources, which covers almost all the dominant unintentional sources of persistent organic pollutants (POPs). The necessity of studying HCBd emissions from those industrial sources lies in the following aspects: Firstly, HCBd emitted from industrial sources dominantly attached on fine particles and be stabilized in the environment and accumulated. Therefore, even though the emission concentrations of HCBd were not very high, the continuous emissions of HCBd from numerous industrial activities can pose continuous and accumulative

risks on the environment and human health. Secondly, there is still a lack of accurate and comprehensive data on HCBD emissions from industrial thermal sources. Field studies on HCBD emissions from industrial sources can derive the emission factors and were essential data for compiling a complete global emission inventory.

References:

[1] Wang, M. et al. Hexachlorobutadiene emissions from typical chemical plants. *Front. Env. Sci. Eng.* **15**, 60 (2021).

Specific comment 4. Can the results in this study explain the ultra-high concentrations of HCBD in Mongolia and the Pacific Islands? The sources lead to those ultra-high concentrations might be different from the investigated sources.

Response: Solving the problem of source identification of POPs has long been a prominent subject in the field of POPs emission research^[1]. However, unlike pollutants with different congener fingerprints from different industrial sources such as dioxins, HCBD has only one congener and cannot differentiate HCBD emissions originating from different sources. Therefore, the results of this study cannot provide direct evidence for the high-concentration HCBD previously reported in Mongolia and the Pacific Islands. The present study is the first to document eight previously unidentified sources of hexachlorobutadiene emissions, including SCu. This study provides theoretical guidance for new sources of atmospheric HCBD and provides data supporting for industrial HCBD pollution.

References:

[1] Zhang, H. et al. A review of sources, environmental occurrences and human exposure risks of hexachlorobutadiene and its association with some other chlorinated organics. *Environmental Pollution* **253**, 831–840 (2019).

Specific comment 5. What is the mechanism of high formation of HCBD in secondary copper smelting, electric arc furnace and hazardous waste incinerations? Or can the reasons leading to high concentrations in those three sources be explained?

Response: There are three important factors influencing unintentional formation of HCBD during thermal processes, including contents of chlorine and organic carbon in the raw materials, the reaction temperature and catalyzing effects of metal compounds^[1]. The secondary copper smelting, electric arc furnace and hazardous waste incinerations of highest HCBD formations are all operated with sufficiently high chlorine- and organic carbon- contents in the raw materials. The secondary copper smelting, electric arc furnace industries uses scrap copper, scrap steel and hazardous wastes as raw materials respectively, which

contains organic residues such as rubber, cable wrapping, plastics and polyvinyl chloride (PVC), contributing to relatively higher HCBD formation in these two industries. Hazardous waste incineration industry disposes hazardous wastes including waste liquids, gases, and residues generated during industrial production processes, as well as medical waste and urban garbage. These hazardous materials can also provide precursors for HCBD formation, such as carbon tetrachloride, acetylene, benzene, et al., contributing to higher formation of HCBD during hazardous material incineration processes. Secondly, temperature is also an important influencing factor for HCBD formation and degradation. There are two possible formation paths for HCBD in industrial thermal processes. One is through precursor synthesis, that is, chlorinated organic precursors transformed to HCBD at 150 ~ 400 °C. The other is that non-chlorinated organic precursors reacted with chlorides at 600 ~ 900 °C to generate HCBD. The cooling down zone of stack gas during industrial processes are at a wide temperature range decreased from about 1200 °C to 200 °C, which inevitably contributed to HCBD generation. In addition, metal oxides/chlorides act as catalysts and have the ability to lower the energy barriers of reactions, which can facilitate the formation of persistent organic pollutants during thermal-related industrial processes. Therefore, the secondary copper smelting and electric arc furnace are both equipped with sufficient precursors and metal catalysts, so it's not surprised that higher HCBD were formed during these two industrial sources. Relevant discussions about the reasons for higher HCBD concentrations from secondary copper smelting, electric arc furnace and hazardous waste incinerations has been added in the revised manuscript (*Line 142-148, Page 6-7*), which were also shown below.

“This high concentration of HCBD in EAF steelmaking could be caused by the presence of organic residues in scrap raw materials, such as rubber, cable wrapping, plastics, and polyvinyl chloride, which provide additional sources of chlorine for the unintentional formation and emission of HCBD^[2]. Yang et al. speculated that differences in the chlorine contents of raw materials and metal catalysts are important contributors to differences in the emissions of chlorinated organic pollutants^[3].”

Reference:

[1]. B. R. Stanmore, *The formation of dioxins in combustion systems*, *Combustion and Flame* 2004 Vol. 136 Issue 3 Pages 398-427.

[2] Pan, S. Y. et al. *Characteristics of PCDD/Fs in PM_{2.5} from emission stacks and the nearby ambient air in Taiwan*. *Sci Rep* 11, 8093 (2021).

[3] Yang, L. et al. *Occurrence of chlorinated and brominated polycyclic aromatic hydrocarbons from electric arc furnace for steelmaking*. *Environmental Pollution* 294, 118663 (2022).

Specific comment 6. ^{13}C -labelled HCBD was used for internal standard of instrumental analysis. Can a GC/MS spectrum be provided? SI

Response: We have provided a GC/MS spectrum as shown below according to the reviewer's comments.

Figure R5. GC/MS Spectra of HCBD and its isotopic internal standard

(Supplementary Figure 4)

Specific comment 7. This section should not involve descriptions of the methods, which should be elaborated in the corresponding section of the Methods.

Response: We have moved the description about the methods to the "Methods" section according to the reviewer's comments (**Line 402-418, Page 18**).

Specific comment 8. It is recommended that this section be augmented with the practical and policy implications of the findings of this study and an outlook for future research

Response: Thanks for your comment. We have made supplements about the practical and policy

implications and an outlook for future research in the revised manuscript according to the reviewer's comment in the revised manuscript (Line 295-306, Page 13), which were also shown below.

“Our results show that these 12 thermal industries are important sources of HCBd, but the existing policies lack the regulation of HCBd in thermal industrial processes. In China, HCBd appears to be primarily limited to industries such as petroleum refining. Therefore, it is imperative to investigate the formation mechanism across diverse industrial processes and strengthen emission limits pertaining to HCBd from various industrial sources. Twelve types of industrial sources were analyzed for HCBd release. It provides a data basis for the formulation of HCBd emission inventory and a scientific basis for global control and emission reduction of HCBd.

This study identified the priority control industries SCu, EAF and COP for high HCBd emissions. Optimizing the technical processes of the above industries and focusing on strengthening terminal emission control can effectively reduce the global emissions of HCBd and the potential health risks caused by HCBd.”

Reviewer #3:

General Comments:

This study delved into HCBd emissions from 121 industrial plants operating at full scale, belonging to 12 big industries. The collected industrial samples were rare and valuable. The extensive data in this study could provide excellent data support for HCBd emission factors and inventory establishment. The study proposed prioritizing the control of HCBd emissions in industries such as secondary copper smelting, electric arc furnace steelmaking, and hazardous waste incineration. The results can be beneficial for international convention implementation. This study is well designed and could improve the understanding of HCBd source emissions. Moreover, the discussion about the impact of environmental behaviors such as dry deposition and oxidation reactions on HCBd transportation is interesting and could provide important supporting information for assessing the potential risks of HCBd exposure. Minor revision is recommended before it could be accepted for publication. Detailed suggestions were shown below:

Response: We greatly appreciate the reviewer's very positive comments on the novelty and significance of this big field investigation about HCBd emissions. We have carefully revised our manuscript based on

the reviewer's comments. Detailed responses to the comments are provided below.

Comment 1. The formation mechanisms of HCBD, as well as the influence of factors such as processing temperature and degradation temperature, need to be included and discussed. The different HCBD concentrations in different industrial PM from various sources should be explained from the viewpoint of formation mechanisms and factors.

Response: Thanks for the reviewer's suggestions. We have provided responses to the Reviewer #2's specific comments 5 (Results and Discussion section). We provided the responses to explain the formation of high concentrations of HCBD below again.

There are three critical factors influencing unintentional formation of HCBD during industrial thermal processes, including contents of chlorine and organic carbon in the raw materials, the reaction temperature and catalyzing effects of metal compounds^[1]. Firstly, the secondary copper smelting, electric arc furnace and hazardous waste incinerations of highest HCBD formations are all operated with sufficient chlorine- and carbon- containing raw materials. The secondary copper smelting, electric arc furnace industries uses scrap copper, scrap steel and hazardous wastes as raw materials respectively, which contains organic residues such as rubber, cable wrapping, plastics and polyvinyl chloride (PVC), contributing to relatively higher HCBD formation in these two industries. Hazardous waste incineration industry disposes hazardous wastes including waste liquids, gases, and residues generated during industrial production processes, as well as medical waste and urban garbage. These hazardous materials can also provide precursors for HCBD formation, such as carbon tetrachloride, acetylene, benzene, et al., contributing to higher formation of HCBD during hazardous material incineration processes. Therefore, differences in the chlorine and organic carbon contents of raw materials is an important contributor to differences in the emissions of HCBD from those 12 industrial sources. Secondly, temperature is also an important influence factor for HCBD formation and degradation. There are two possible formation paths for HCBD in industrial thermal processes. One is through precursor synthesis, that is, chlorinated organic precursors transformed to HCBD at 150 ~ 400 °C. The other is that non-chlorinated organic precursors reacted with chlorides at 600 ~ 900 °C to generate HCBD. The cooling down zone of stack gas during industrial processes are at a wide temperature range decreased from about 1200 °C to 200 °C, which inevitably contributed to HCBD generation. Thirdly, metal oxides/chlorides act as catalysts and have the ability to lower the energy barriers of reactions, which can facilitate the formation of persistent organic pollutants during thermal-related industrial processes. Therefore, the secondary copper smelting and iron

ore sintering are equipped with both sufficient precursors and metal catalysts, so it's not surprised that higher HCBD were formed during these two industrial sources. Relevant discussions of critical influence factors for HCBD generations and the varied HCBD concentrations from different industries has been added in the revised manuscript (Line 142-148, Page 6-7), which were also shown below.

“This high concentration of HCBD in EAF steelmaking could be caused by the presence of organic residues in scrap raw materials, such as rubber, cable wrapping, plastics, and polyvinyl chloride, which provide additional sources of chlorine for the unintentional formation and emission of HCBD^[2]. Yang et al. speculated that differences in the chlorine contents of raw materials and metal catalysts are important contributors to differences in the emissions of chlorinated organic pollutants^[3].”

Reference:

[1]. B. R. Stanmore, *The formation of dioxins in combustion systems*, *Combustion and Flame* 2004 Vol. 136 Issue 3 Pages 398-427.

[2] Pan, S. Y. et al. *Characteristics of PCDD/Fs in PM2.5 from emission stacks and the nearby ambient air in Taiwan*. *Sci Rep* 11, 8093 (2021).

[3] Yang, L. et al. *Occurrence of chlorinated and brominated polycyclic aromatic hydrocarbons from electric arc furnace for steelmaking*. *Environmental Pollution* 294, 118663 (2022).

Comment 2. HCBD is a kind of byproducts from chemical manufacturing industries. The emission concentrations and emission factors of HCBD from chemical manufacturing industries should be further added and discussed. HCBD emissions from typical chemical manufacturing processes with the investigated sources should be compared.

Response: Thank you for your comment. HCBD has no known natural sources. Production and use of HCBD as chemical have been stopped because of its ecotoxicity. Current emissions of HCBD primarily occur as unintentional releases during industrial activities.

The synthesis of various chlorinated hydrocarbons is usually carried out at high temperatures, resulting in the formation of hexachlorobutadiene as a by-product during the manufacturing process of certain chlorinated organic compounds. Chlor-alkali plants and other chemical plants producing trichloroethylene (TCE), tetrachloroethylene (TeCE), carbon tetrachloride (CTC), chloromethane, Hexachlorocyclopentadiene (HCCP) are the typical sources of HCBD emission (**Table R5**)^[1-2].

However, UNEP have not yet released available emission factors of HCBD, and the existing research

on the emission concentration of HCBd in the chemical manufacturing industry is not extensive. Some studies have used the concentration of HCBd in primary products as the proportion of by-products in chemical products^[2], but the concentration data (1991) are very old. Our previous studies have detected the HCBd content in the final products of chemical plants in recent years. While HCBd emissions from this part decrease with the phase-out or strict control of the manufacture and use of TCE, TeCE and CTC in some other nations. The 12 industrial thermal sources investigated in this study exhibit higher yields and a broader distribution across various countries. In terms of emission methods, not all HCBd generated as by-products in the chemical manufacturing industry is directly released^[1]; instead, 40% is incinerated as hazardous waste, 20% undergoes fractionation and recovery, and only a small proportion is discharged directly into the environment. The findings from the investigation of 12 thermal industries indicate that HCBd will be directly emitted into the atmosphere along with PM release, leading to extensive environmental pollution. Therefore, it is imperative to prioritize the control of HCBd emissions from industrial thermal sources. Relevant discussion on HCBd from chemical manufacturing processes has been added in the revised manuscript (Line65-79, Page 4), which were also shown below.

“Currently, a few studies have reported the release of HCBd from chemical production processes that use chlorine and waste disposal^{16,17}. Our previous study has also investigated the occurrence of HCBd in products and bottom liquid of chlorobenzene, trichloroethylene, and tetrachloroethylene chemical manufacturing plants (Supplementary Table 1)¹⁶. HCBd concentrations in the bottom liquid samples contributed 24%–99% of the total HCBd formed in the chemical production plants. The bottom liquid was disposed of as hazardous waste by incineration¹⁸. Therefore, a proportion of HCBd from commercial chemical manufacturing processes would finally enter into environment by the unintentional releases from incinerations of bottom liquid. However, current HCBd emission data from industrial sources were very limited. It has also been found that HCBd occurred in industrial fine particulate matter (i-PM) from waste incineration facilities in East China¹⁹.”

Table R5. HCBd concentration in the raw product from different chemical industries

(Supplementary Table 1)

Items	Process	Concentration (ng/mL)	Reference	Time
CTC	Methane method	5.00×10^{-2}	UNEP, 2013	1991 ^[3]

	Methane method (optimized)	5.00×10^{-3}	UNEP, 2013	1991 ^[3]
	Methanol method	8.17×10^{-5}	Zhang, 2015	2015 ^[2]
	Methanol method	3.96×10^{-7}	Wang, 2021	2021 ^[1]
HCCP	Cyclopentadiene method	1.11×10^{-2}	UNEP, 2013	1991 ^[3]
PCE	Acetylene method	4.00×10^{-3}	Beijing Normal University, 2014	2014
	CTC method	4.20×10^{-3}	Beijing Normal University, 2014	2014
	Acetylene method	7.60×10^{-8}	Wang, 2021	2021 ^[1]
	CTC method	3.96×10^{-4}	Wang, 2021	2021 ^[1]
TCE	Acetylene method	4.00×10^{-3}	UNEP, 2013	1991 ^[3]
	Acetylene method	7.60×10^{-8}	Wang, 2021	2021 ^[1]

References:

- [1] Wang, M. et al. Hexachlorobutadiene emissions from typical chemical plants. *Front. Env. Sci. Eng.* **15**, 60 (2021).
- [2] Zhang, L., Yang, W., Zhang, L. & Li, X. Highly chlorinated unintentionally produced persistent organic pollutants generated during the methanol-based production of chlorinated methanes: A case study in China. *Chemosphere* **133**, 1–5 (2015).
- [3] UNEP. Risk management evaluation on hexachlorobutadiene. <https://pops.int> (2013).

Comment 3. Particle matter (PM) released from industrial emissions is a crucial factor influencing the occurrence and human exposure of HCBD in the environment. However, the manuscript lacks detailed descriptions of the characteristics of industrial PM from various industries. Characterizing and clarifying the PM from 12 different industrial sources is essential, as it may affect their transportation behavior in the atmosphere. Furthermore, will the distinct characteristics of i-PM from diverse industrial sources impact the dry deposition behavior?

Response: Thank you for your comment. In this study, HCBD primarily associates with particulate matter and is emitted into the atmosphere. The characterization of the properties of particulate matter size is important. In our previous study, we conducted a scanning electron microscope (SEM) analysis to examine the fundamental characteristics of these industrial particle matters, revealing their predominant size to be below 2.5 μm . The SEM results have been provided in the materials to *Nature Communications* with DOI number 10.1038/s41467-023-39491-5. The SEM photos (**Figure R1**) have also provided again in the

response to Reviewer #2's specific comments (Introduction section). In this study, the variation in i-PM characteristics is not expected to result in significant differences in the dry deposition of HCBD.

Comment 4. In line 180-182, why compared HCBD EF from coking processes with that from pentachloroene and hexachlorobenzene? Further statement is suggested.

Response: We compared HCBD EF from coking industries with that of pentachloroene and hexachlorobenzene for the reason that they all belong to persistent organic pollutants (POPs) with similar physicochemical properties covered by the Stockholm Convention. HCBD, pentachloroene and hexachlorobenzene can be unintentionally produced during industrial thermal sources. In the previous study, coking industries were considered to be sources of unintentional POPs. The emission factors of pentachlorobenzene and hexachlorobenzene from coking industries were relatively lower compared to that from other industries^[1]. However, we found that the coking industries had a high HCBD EF. The EF of HCBD from coking industry was one order of magnitude higher than the EFs for pentachlorobenzene and hexachlorobenzene. Therefore, HCBD emissions from coking industry should be emphasized in spite of low pentachlorobenzene and hexachlorobenzene emissions.

Reference:

1. Gong, W., Fiedler, H., Liu, X., Wang, B. & Yu, G. Emission factors of unintentional HCB and PeCBz and their correlation with PCDD/PCDF. *Environmental Pollution* **230**, 516–522 (2017).

Comment 5. It is interesting to evaluate the impact of dry deposition on the transportation and persistence of HCBD. Please explain why dry deposition has no obvious effect on HCBD loss in the atmosphere. Were the consumption pathways (dry deposition, oxidation reactions, long-range atmospheric transport [LART]) of HCBD on i-PM different for different industrial sources (Figure 5)?

Response: Thank you for your comment. In this study, the contribution of HCBD transport and retention was evaluated by calculating the dry deposition rate of HCBD near the plant and the rate of participation in the oxidation reaction. Similar to the results of Yang et al.^[1], the proportion of dry deposition of HCBD calculated in this study is also low, indicating that the dry deposition process has no significant effect on the loss of HCBD in the atmosphere.

Considering the particle-gas phase interconversion of HCBD in the atmospheric environment, and to mitigate the influence of complex factors such as temperature and humidity, we incorporated experimental data from Sabin et al^[2]. into our calculation model to accurately quantify the dry deposition behavior of

HCBD in real environment.

References:

[1] Yang, M. et al. *Quantifying Concentrations and Emissions of Hexachlorobutadiene - A New Atmospheric Persistent Organic Pollutant in northern China. Environmental Research* 114139 (2022).

[2] Sabin, L. D. et al. *Exchange of Polycyclic Aromatic Hydrocarbons Among the Atmosphere, Water, and Sediment in Coastal Embayments of Southern California, Usa. Environ. Toxicol. Chem.* 29, 265–274 (2010).

Comment 6. In Line 253, the authors mentioned that "higher emissions of HCBD in the Global South than in the Global North indicate higher HCBD exposure for individuals in the Global South compared to those in the Global North." The lifetimes of HCBD might be relatively shorter in the Global South compared to the Global North. This should be mentioned when assessing the exposure risks in these regions.

Response: Thanks for the reviewer's kind suggestion. HCBD is a persistent organic pollutant whose half-life in the environment is influenced by factors such as temperature, humidity, soil type, and microbial activity. In general, under warm and humid climate conditions, the degradation rate of organic pollutants may accelerate. The climate in the Global South, which is mainly tropical and subtropical, features year-round high temperatures and heavy rainfall. Therefore, the half-life of HCBD in the global South may be shorter than that in the global North. Conversely, the climate in the Global North, which is mainly temperate and Arctic, is characterized by long cold winters and short cool summers. Under those cold and dry conditions, the degradation rate of organic pollutants may slow down, leading to a longer half-life of HCBD in the Global North compared to the global South. We have mentioned the potential influence of climate conditions of Global South and Global North on HCBD half-life when discussing the HCBD exposure risks in these regions in the revised manuscript (as shown in blue text below. Line 290-294, Page 13).

“In addition, considering the completely different climate conditions in the Global South and Global North, the environmental behavior of HCBD, such as dry deposition, oxidation reactions, were different and should be taken into consideration when comprehensively assessing HCBD exposure risks in these regions.”

Reviewer #2 (Remarks to the Author):

All the comments are addressed well. I suggest an acceptance.

Reviewer #3 (Remarks to the Author):

The authors gave a detailed and clear response to the reviewer's comments. After revision, the quality of the manuscript has been significantly improved.